# Increasing global precipitation whiplash due to anthropogenic greenhouse gas emissions

Xuezhi Tan [1,2,4,5] ✉, Xinxin Wu [1,4,5], Zeqin Huang[1], Jianyu Fu[1], Xuejin Tan[1], Simin Deng[1], Yaxin Liu[1], Thian Yew Gan [3] & Bingjun Liu[1]

Precipitation whiplash, including abrupt shifts between wet and dry extremes, can cause large adverse impacts on human and natural systems. Here we quantify observed and projected changes in characteristics of sub-seasonal precipitation whiplash and investigate the role of individual anthropogenic influences on these changes. Results show that the occurrence frequency of global precipitation whiplash is projected to be $2.56 \pm 0.16$ times higher than in 1979–2019 by the end of the 21st Century, with increasingly rapid and intense transitions between two extremes. The most dramatic increases of whiplash show in the polar and monsoon regions. Changes in precipitation whiplash show a much higher percentage change than precipitation totals. In historical simulations, anthropogenic greenhouse gas (GHG) and aerosol emissions have increased and decreased precipitation whiplash occurrences, respectively. By 2079, anthropogenic GHGs are projected to increase $55 \pm 4\%$ of the occurrences risk of precipitation whiplash, which is driven by shifts in circulation patterns conducive to precipitation extremes.

Abrupt shifts in precipitation regimes, also known as precipitation whiplash, from wet extremes to dry extremes or vice versa, can result in cascaded heavy impacts on the ecosystem and human society adversely. For example, vegetation green-up during the wet extremes period could provide sufficient fuel for the worst wildfire that occurred in the following dry extremes period[1–3]. On the other hand, although whiplash from dry extremes to wet extremes is of critical importance to the regional recovery of available water resources, the surface soil dried by dry extremes may form a dense surface crust which favors the generation of the surface runoff and flooding, and concurrent landslides[4] and erosion[5]. Increased precipitation whiplash from drought to flood can also increase riverine nitrogen loads and concentrations in the agricultural lands[6], amplifying negative trends in water quality. A precipitation whiplash event often refers to a dry (wet) extreme immediately following a wet (dry) extreme with no break in the normal precipitation regimes[7]. Decreasing shift time between the two precipitation extremes poses little time for human preparedness

to adapt to the extreme circumstance, thus exacerbating the individual impacts of the drought[8] or pluvial[9].

Precipitation variability is projected to increase in the 21st Century on daily-to-multiyear timescales with distinct regional characteristics[10,11] in a warmer world[12–15], which could further increase the vulnerability of the ecosystem to changes in precipitation and challenge the climate resilience of human society and infrastructure. Previous regional analyses examined transitions and variability of wet and dry extremes on relatively longer (seasonal or annual) timescales using various methods[7,16–23]. Standardized precipitation indices were used to characterize the transition features of regional precipitation regimes on the seasonal scale[16,21,22]. Annual anomalous precipitation dipole events (a drought year followed by a pluvial year[18]) and seasonal precipitation anomalies (a dry winter followed by a wet winter) were analyzed to focus on the transition between long-term dry and wet regimes[7]. Although sub-seasonal abrupt shifts in large-scale atmospheric circulations for regional climate extremes have been analyzed

[1]Center of Water Resources and Environment, School of Civil Engineering, Sun Yat-sen University, Guangzhou 510275, PR China. [2]Southern Marine Science and Engineering Guangdong Laboratory (Zhuhai), Zhuhai 510275, PR China. [3]Department of Civil and Environmental Engineering, University of Alberta, Edmonton, AB T6G 2W2, Canada. [4]These authors contributed equally: Xuezhi Tan, Xinxin Wu. [5]These authors jointly supervised this work: Xuezhi Tan, Xinxin Wu. ✉e-mail: tanxuezhi@mail.sysu.edu.cn

recently[24–26], the sub-seasonal climatology and changes in global precipitation whiplash of rapid and intense wet-dry and dry-wet transitions have yet to be examined. The global investigation of characteristics and timing of precipitation whiplash also remains unclear. Therefore, we propose a metric to measure precipitation whiplash on the sub-seasonal time scale over the globe (Methods) and use the metric to detect changes in precipitation whiplash.

Intensification of the hydrological cycle and subsequent changes in precipitation variability and wet and dry spells stems from the thermodynamic and dynamic responses to global climate change under various forcings involved[27]. Greenhouse gases and aerosol emissions are two primary anthropogenic activities that show warming[28] and cooling[29] effects of climate change, respectively. Because greenhouse gases in the atmosphere are generally well-mixed while aerosols are locally distributed around the emission sources[30,31], regional precipitation changes are complicated by the coupling responses to various forcings. Given that in the coming decades, greenhouse gases will increase while aerosol emissions will decrease significantly worldwide, it is of extreme potential to understand the effects of greenhouse gases and aerosol forcings on precipitation whiplash[32].

Here we detect spatiotemporal changes in observed and projected occurrence frequency, transition duration, and intensity of precipitation whiplash from the gridded datasets, CESM Large Ensemble Community Project (CESM-LENS)[33], CESM1 'all-but-one' experiments (CESM-XLENS)[34], and Coupled Model Intercomparison Project Phase 6 (CMIP6)[35] global climate model (GCM) simulations from 1920–2100 under different forcings. We consider forcings, including the historical (ALL), anthropogenic greenhouse gases (GHG), anthropogenic aerosols (AER), and biomass burning aerosols (BMB), adopted in the state-of-the-art GCMs (Methods). For each ensemble in the dataset, we identify wet (dry) extremes based on the exceedance (deficit) of 30 consecutive days' precipitation above (below) the 90th (10th) climatological percentile of those precipitation events in the current period (1979–2019), respectively. To eliminate the effects of global warming on the persistent wetting or drying, we linearly detrend and normalized precipitation time series with climatological annual cycles removed before identifying precipitation extremes (see Methods). An intra-seasonal rapid transition from a dry (wet) extreme to a wet (dry) extreme is defined as a precipitation whiplash of dry-to-wet (wet-to-dry). Our efforts attempt to disentangle potentially competing influences on changes in precipitation whiplash from regional to global scales and improve the understanding of their future changes.

## Results

### Global climatology of precipitation whiplash

The CESM-LENS and CMIP6 historical ensembles reasonably reproduce the historical climatology of the occurrence frequency of precipitation whiplash (Fig. 1 and Supplementary Figs. 6, 7), which are well within the range of reanalyses, satellite- and ground-based precipitation. Both observed and simulated precipitation show consistent latitudinal differences in the occurrence characteristics of precipitation whiplash (Fig. 1a–f). The Tropics and the mid-latitudes (-30–60°) in both the Northern and Southern Hemisphere show extremely low and high occurrence frequency, respectively, which is concurrent to the low and high precipitation seasonality caused by different dominant circulation conditions associated with precipitation over these two regions. Precipitation whiplash occurs less than 0.2 times per year at the low latitudes, while it occurs -0.5 times per year over the mid- and high-latitudes. Different from the occurrence frequency, the whiplash transition duration is shortest in the mid- and low-latitudes (-10–30°) in both Hemispheres, indicating a relatively more rapid precipitation transition in these regions. The transition intensity is relatively large in subtropical subsidence regions but low in the polar regions (Fig. 1e, f).

The average timing of precipitation whiplash shows distinct spatial patterns (Fig. 1g, h). Most precipitation whiplash occurs in June-October (November-March) on the eastern (western) sides of the Northern Hemisphere's continents, including Asia (Europe) and the eastern (western) side of North America. The timing of dry-to-wet whiplash is generally 1–2 months later than that of wet-to-dry whiplash. In the land areas of the Southern Hemisphere, whiplash occurs mainly in December-January. Oceans are characterized by frequent whiplash occurrences in September-March and March-September for the Northern and Southern Hemisphere, respectively. The Antarctic shows similar timing of precipitation whiplash as those of the ocean in the Southern Hemisphere. There is no significant change in the timing of the occurrence in the mid- and low-latitudes in both Hemispheres from the historical period (Fig. 1g, h) to the future period (Supplementary Fig. 8). However, the average timing of occurrence in the high-latitudes changes noticeably, with dry-to-wet events in the historical period occurring earlier than those in the current period in the Arctic Ocean and wet-to-dry events occurring later. The high latitudes of the Southern Hemisphere show an opposite pattern of those changes in the Northern Hemisphere.

### Projected changes in precipitation whiplash

To investigate the response of whiplash to climate warming, here we take the current period (1979–2019) as a baseline and quantify historical and future changes in the event frequency, transition duration, and intensity of precipitation whiplash until the end of the 21st Century (2060–2099) under the RCP8.5 emissions scenario. The agreement of projected changes in the three characteristics of precipitation whiplash across the global and land mean of observations, CESM-LENS (Fig. 2 and Supplementary Fig. 9), and CMIP6 models (Supplementary Figs. 10, 11) indicates the robustness of the results. Since the late 1990s, both observations and simulations show increases in the global and land mean frequency of whiplash events (Fig. 2b and Supplementary Figs. 9–11b). The land-only datasets (CHIRPS, GPCC, and REGEN) and land simulations indicate stronger changes in whiplash characteristics over land than the global average.

Globally, most regions show statistically significant and robust changes in characteristics of precipitation whiplash, suggesting that global warming exerts substantial impacts on precipitation whiplash, despite strong spatial heterogeneity (Fig. 2a, c, e and Supplementary Figs. 9–11). Projected changes in occurrences of dry-to-wet (Fig. 2b, d, f and Supplementary Fig. 9) and wet-to-dry (Fig. 3b, d, f and Supplementary Fig. 10) events share similar features in both the global and land mean trends, with trends continuing to the end of the 21st Century. The global frequency of dry-to-wet (wet-to-dry) events is projected to increase by 78 ± 8%, which implies a total increase in whiplash events by 156 ± 16% (2.56 ± 0.16 times) of that in the current period. Land areas will experience a more intense response, with a total increase of 243 ± 22% (3.43 ± 0.22 times) of that in the current period in both types of whiplash. This phenomenon may result from an intensification of the hydrological cycle as the climate warms and the globe thus becomes wetter and more variable simultaneously[13].

The temporal evolution of the signal-to-noise ratio allows for estimating the timing of the emergence of external forcing. The decadal impact of external forcing on whiplash occurrence is largely uncertain in the current period and before, as it is weaker than the internal variability component (Supplementary Figs. 12, 13). However, both CESM-LENS and CMIP6 simulations show that the global and land mean forcing responses of dry-to-wet (wet-to-dry) whiplash begin to be larger than the internal variability around 2028 and 2017 (2033 and 2017), and show significant and continuous increases in the occurrence frequency of precipitation whiplash. The result is robust, as only two members of each ensemble are required to detect signals from noise and thus obtain significant changes in the frequency of whiplash occurrence by the 2050s (2040 s) onwards. Most of the globe, except

for the mid- and low-latitude oceans, can be diagnosed with a signal of anthropogenic changes in precipitation whiplash in the 21st Century (Supplementary Figs. 12, 13a, b), which provides a base for our next step to detect responses from the single forcing of anthropogenic activities. The increasing occurrence frequency of future precipitation whiplash events is accompanied by shorter transition durations (−10 ± 1%), implying increasingly sharp transitions which leave less time for decision-makers to react (Fig. 2c, d and Supplementary

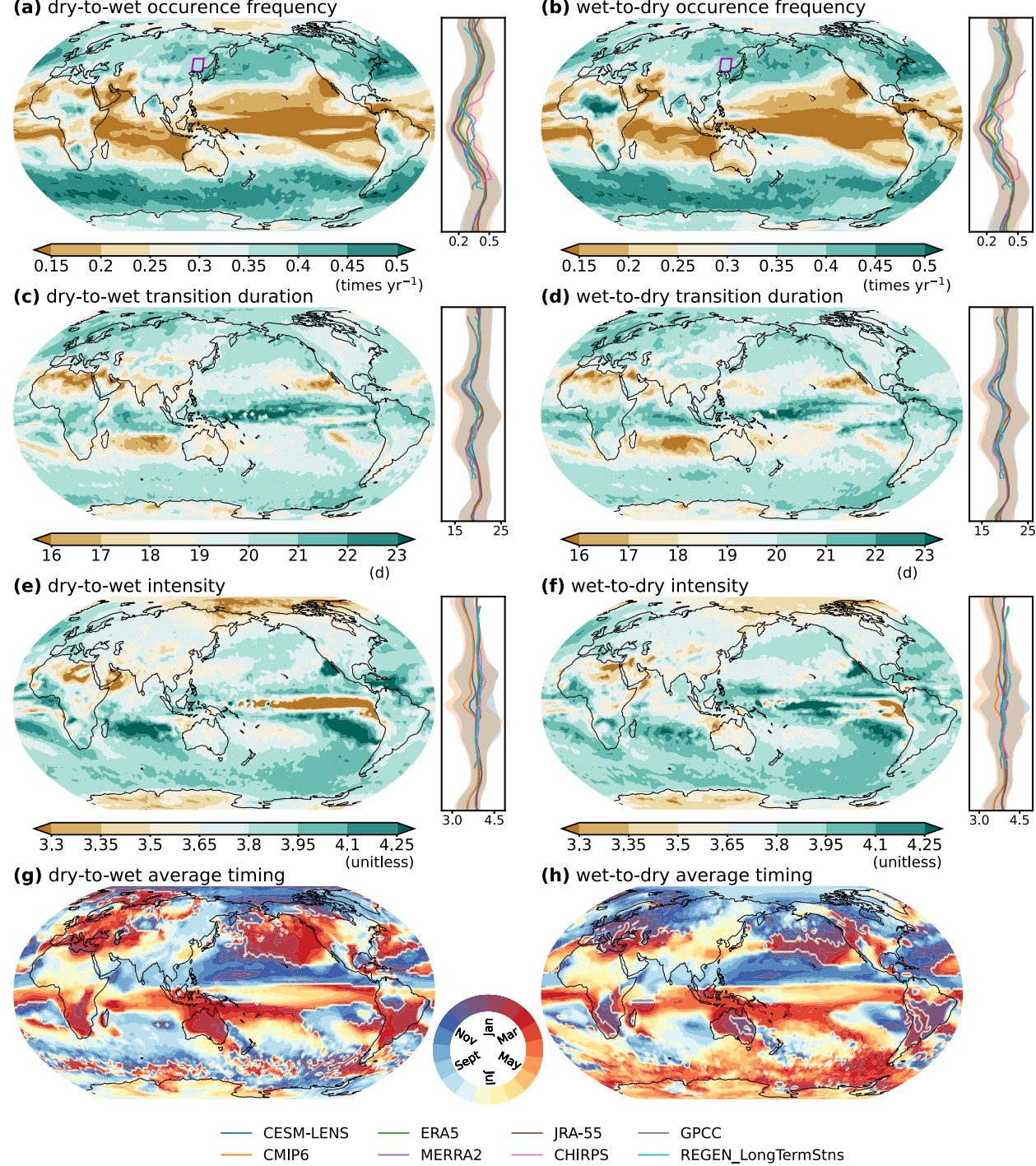

**Fig. 1 | Climatology of occurrence frequency, transition duration, intensity, and average timing of precipitation whiplash.** Maps show the ensemble mean climatology of **a**, **b** occurrence frequency (unit: times year⁻¹), **c**, **d** transition duration (unit: days), **e**, **f** intensity (unitless), and **g**, **h** average occurrence timing of dry-to-wet (**a**, **c**, **e**, **g**) and wet-to-dry whiplash (**b**, **d**, **f**, **h**) over the current period (1979–2019) in the CESM-LENS ensemble. The lines accompanying the maps **a**-**f** show the zonal means of the climatology of the corresponding characteristic derived from the CESM-LENS, CMIP6, and six precipitation datasets. Note that CHIRPS only have a coverage of 50°S–50°N land area and GPCC and REGEN_-LongTermStns have a coverage of land only. The orange (blue) shaded area shows the spread of zonal mean values in 90% of the CESM-LENS (CMIP6) ensemble. The purple boxes in maps **a**, **b** indicate the northeastern China region used for the analysis of large-scale circulation background during precipitation whiplash (Fig. 6).

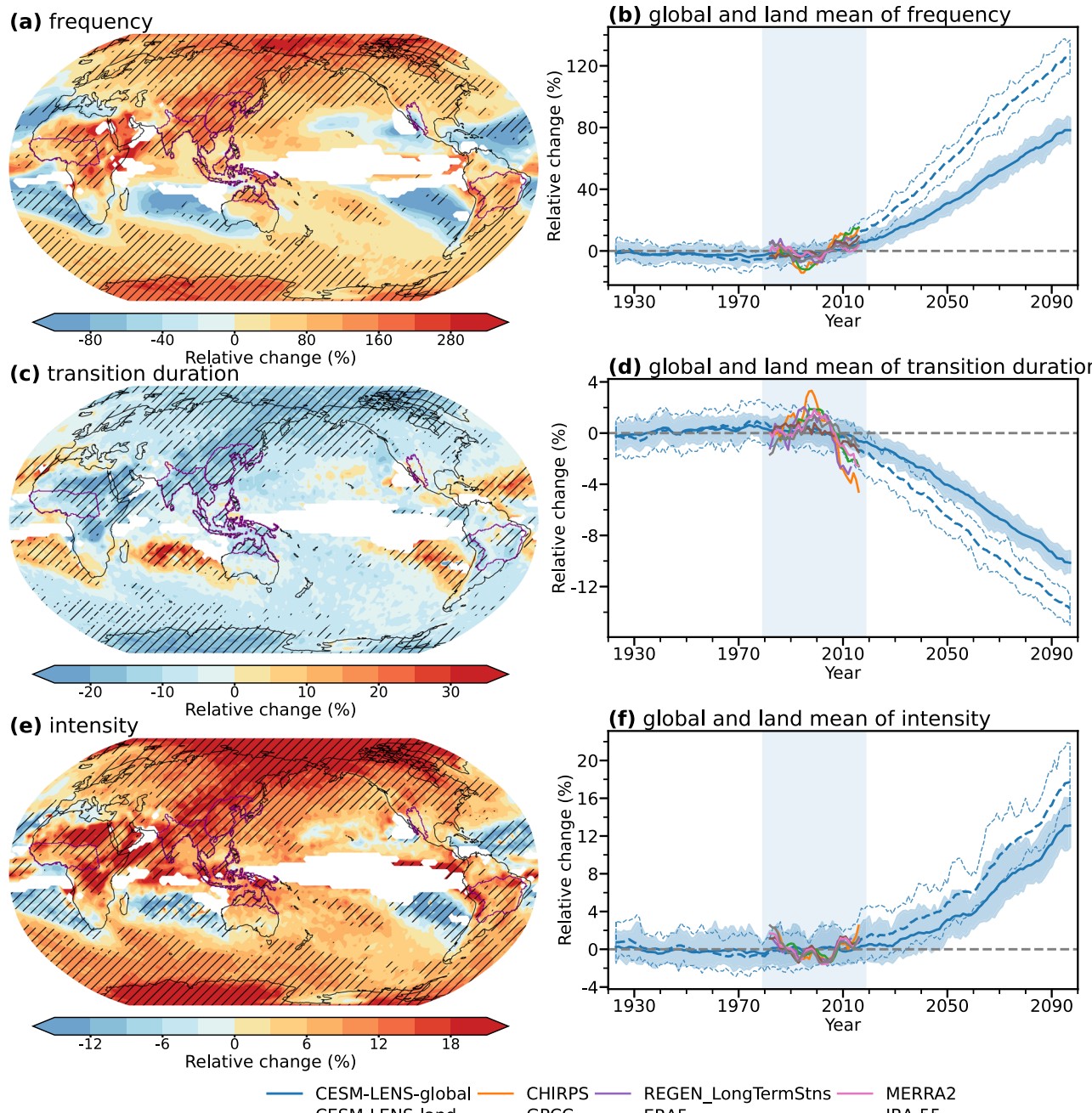

**Fig. 2 | Projected relative changes in the occurrence characteristics of dry-to-wet whiplash over 1921–2099. a**, **c**, **e** show projected relative change (%) in **a** occurrence frequency, **c** transition duration, and **e** intensity of dry-to-wet whiplash in the last four decades of the 21st Century (2060–2099) under the RCP8.5 forcing relative to the current period (1979–2019). Hatching shows that more than 90% of the ensemble members agree on the changes in occurrence characteristics across the 40-member CESM-LENS ensemble. Regions showing fewer than five whiplash events in total for the 40 ensemble members in the current period are masked. Global mean value of global area-weighted average relative changes (%) in **b** occurrence frequency, **d** transition duration, and **f** intensity of dry-to-wet whiplash (dashed line for land mean and solid line) derived from the CESM-LENS ensemble. Time series are moving averaged over 5-year intervals. The shaded area shows the spread of values in 90% of the ensemble. The baseline (gray dashed horizontal line) for the relative changes in the occurrence characteristics of whiplash is the respective mean value over the current period (1979–2019). Note that CHIRPS has a coverage of the 50°S–50°N land area and GPCC and REGEN_LongTermStns have a coverage of land only.

Figs. 9–11), and increasing intensity (13 ± 3%), implying a growing severity of events (Fig. 2e, f and Supplementary Figs. 9–11). The spatial patterns of changes in the occurrence frequency of the two-type whiplash are similar, with most regions tending to be more volatile by the end of the 21st Century. The most dramatic increases occur in the polar and monsoon regions.

We, therefore, pay particular attention to the six monsoon regions, where precipitation changes exert a substantial socioeconomic impact on 2/3 of the global population and whose extreme precipitation is highly sensitive to global warming[36]. We examined how the frequency of this two-type whiplash in the six monsoon regions will change by the late 21st Century relative to the current period (1979–2019) (Fig. 3). The CMIP6 simulation results show a high agreement with those of the CESM-LENS in most regions. The monsoon region on the west of the Pacific is the most affected and will experience substantial increases in whiplash events in the future.

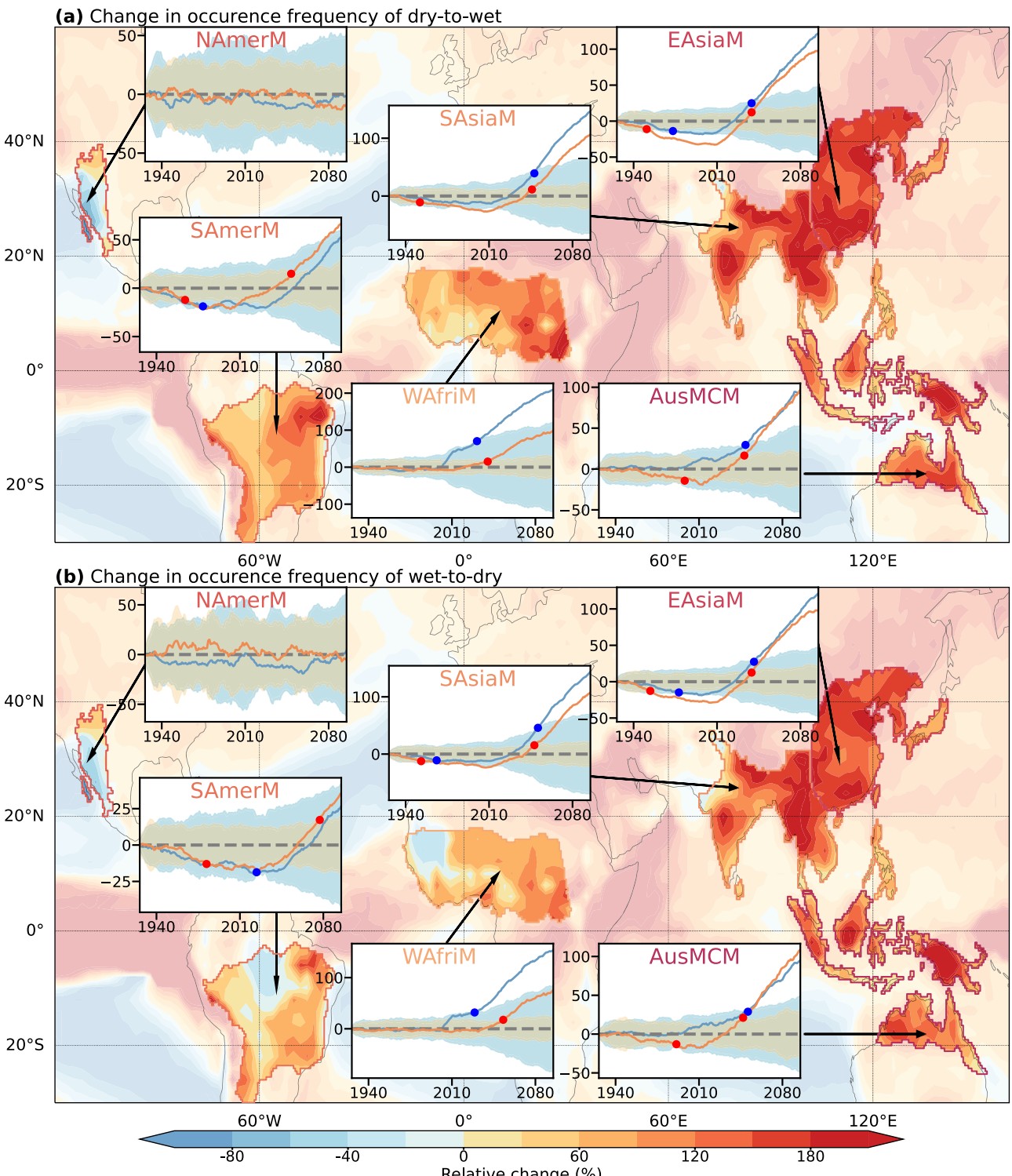

**Fig. 3 | Projected relative changes in the regional occurrence characteristics of precipitation whiplash.** Relative changes (%) in the occurrence frequency of dry-to-wet whiplash (**a**) and wet-to-dry whiplash (**b**) events at the last four decades of the 21st Century (2060–2100) under the RCP8.5 forcing relative to the recent four decades (1979–2019) over the six monsoon regions including North American monsoon (NAmerM), South American monsoon (SAmerM), West African monsoon (WAfriM), South and Southeast Asian monsoon (SAsiaM), East Asian monsoon (EAsiaM), and Australian-Maritime Continent monsoon (AusMCM) regions. Plots on maps indicate the area-weighted average changes in the occurrence frequency of precipitation whiplash in all monsoon regions derived from the ensemble mean of CMIP6 and CESM-LENS simulations. Data were smoothed over 10-year intervals. The regional mean forced response (lines) and the uncertainties standard deviation (shadings) in CESM-LENS (orange) and CMIP6 (blue) are represented. Orange (blue) circles are the first year when the forced response is greater than the uncertainties, i.e., the absolute value of signal-to-noise ratio (S/N) is greater than or equal to 1, in CESM-LENS (CMIP6).

**(a)** dry-to-wet

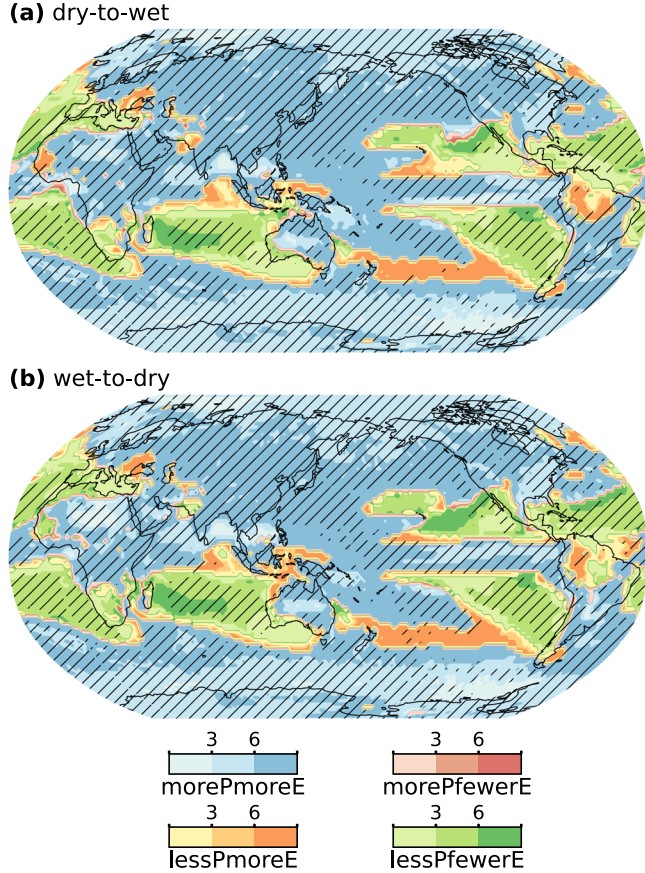

**(b)** wet-to-dry

3   6                    3   6
morePmoreE              morePfewerE
3   6                    3   6
lessPmoreE              lessPfewerE

**Fig. 4 | Concurrent changes in precipitation totals and precipitation whiplash occurrences.** Maps show the ratio of relative changes in ensemble mean occurrence frequency of dry-to-wet whiplash (**a**) and wet-to-dry whiplash (**b**) to the contemporaneous relative change in precipitation totals in the CESM-LENS ensemble. Relative changes (%) are the changes in characteristics at the last four decades (2060–2100) of the 21st Century under the RCP8.5 forcing relative to the recent four decades (1979–2019). For each regime, the darker the color is, the larger the magnitude of change in the whiplash occurrence frequency concurrent with those changes in precipitation totals. Hatching shows more than 90% of the ensemble members agree on the concurrent change regime category.

Almost all grid cells in EAsiaM, SAsiaM, and AusMCM show a consistent increase in the occurrence frequency of whiplash. Whiplash events in EAsiaM (SAsiaM) will increase 196% (214%), consisting of 98% (106%) of dry-to-wet and 98% (107%) of wet-to-dry whiplash events, and signals of anthropogenic warming are projected to show in 2039 and 2047 for EAsiaM and SAsiaM, respectively. Notably, these two regions show an external forcing signal in the 1950s when whiplash events became less frequent. Regions of AusMCM will also show more whiplash (206%, consisting of 97% and 109% for dry-to-wet and wet-to-dry whiplash events, respectively). WAfirM is projected to show more than 50% increases in whiplash occurrences, although CESM-LENS and CMIP6 simulations show a large projected occurrence difference. Whiplash events in SAmerM only increase by 25%. There is almost no change in whiplash occurrences over NAmerM. The spatial distributions of decreasing transition duration (Supplementary Fig. 14) and increasing intensity (Supplementary Fig. 15) of the two-type whiplash events indicate that whiplash events become sharper in all monsoon regions except for NAmerM.

We also analyze possible relations between changes in precipitation whiplash and precipitation totals (Fig. 4). Globally, changes in dry-to-wet (wet-to-dry) whiplash and precipitation totals are positively correlated in 91% (90%) of the regions. This illustrates the strong correlation between precipitation whiplash and precipitation totals. In 76% (75%) of the globe, the occurrences of dry-to-wet (wet-to-dry) events increase with increased precipitation totals, and the percentage increases in whiplash occurrences are higher than that of precipitation totals. The percentage increases in the occurrences of whiplash in 2/3 of these regions are more than six times that of precipitation totals, showing a rapid rate of increase in the occurrence of precipitation whiplash. This is more common over land (88 and 87% for dry-to-wet and wet-to-dry whiplash, respectively), indicating that most of the land will experience a wetter and more volatile hydroclimate. Dry-to-wet (wet-to-dry) whiplash is projected to decrease over 12% (11%) of the land. Nearly 50% of global regions showing decreasing precipitation will become more frequent with precipitation whiplash events, mainly in marine regions at the margin between the reduced and increased precipitation, and land regions, including southeastern Europe, the western North American monsoon region, and the Amazon.

## Anthropogenic impacts on precipitation whiplash

Given that most regions are projected to experience anthropogenic changes in precipitation whiplash after 2020 (Supplementary Figs. 12, 13), we focus on the average relative influence of different anthropogenic forcings up to 2079 (Fig. 5). The whiplash frequency of the ensemble that fixes GHGs in 1920 (XGHG) remains unchanged until the future, suggesting that if GHG emissions do not increase, the simulated whiplash will not change substantially in the future. Although the effects of GHGs on precipitation whiplash events were relatively small before the mid-Twentieth Century, their enlarging effects on whiplash are projected to amplify gradually thereafter. By 2028, GHGs will increase the risk of both types of whiplash by $13 \pm 2\%$ on average. The impact of GHGs will persistently enhance precipitation volatility in the future and is projected to increase the risk of whiplash by $55 \pm 4\%$ ($59 \pm 4\%$) over the globe (land areas) by 2079 (Fig. 5a, d). GHGs contribute $87 \pm 4\%$ of changes in the occurrence frequency of whiplash in the future period (2040–2079) relative to the current period (1979–2019). The region of the relatively great influence of GHGs is largely consistent with the region of high signal-to-noise ratio we identified, implying that GHGs are the main external forcing of changes in precipitation whiplash (Fig. 5c, f). GHGs amplify the increase in whiplash in the mid- and high-latitudes and monsoon regions, except the NAmerM regions. By 2079, in the regions of AusMCM, EAsiaM, and SAsiaM, more than 50% of average increases in whiplash risk will be amplified by GHG emissions, with similar magnitudes for dry-to-wet and wet-to-dry whiplash (Supplementary Fig. 16). Most notably, during 2040–2079, projected risk of whiplash in polar regions will increase >1.2 times (Fig. 5c, f). By 2079, GHGs are projected to increase the risk of global mean whiplash transition duration shortening by $4 \pm 1\%$ (even though not significant in most of the regions, Supplementary Fig. 17). As for whiplash intensity, GHGs are projected to amplify the risk of global mean whiplash intensity by $7 \pm 1\%$ (mainly significant in the polar regions and Asia, Supplementary Fig. 18). $40 \pm 5\%$ of the reduction in transition duration of whiplash in the future period relative to the current period is contributed by GHGs, and $80 \pm 12\%$ of the intensity enhancement is contributed by GHGs. The results suggest that GHGs are projected to increase the occurrence frequency of precipitation whiplash and facilitate more violent transitions during whiplash events in many economically and demographically important regions of the globe by the end of the 21st Century.

AER and BMB forcings can magnify or offset increases in GHG-driven extreme precipitation events, despite their spatially heterogeneous effects. AER globally drives the opposite changes of GHGs for precipitation whiplash events, and exerts counterbalancing influences on GHGs in southeastern Asia, eastern and southern Africa, and Arctic regions from 1921–2028 (Supplementary Fig. 19). The persistent suppression of East Asian circulation[37] by the AER forcing over the recent

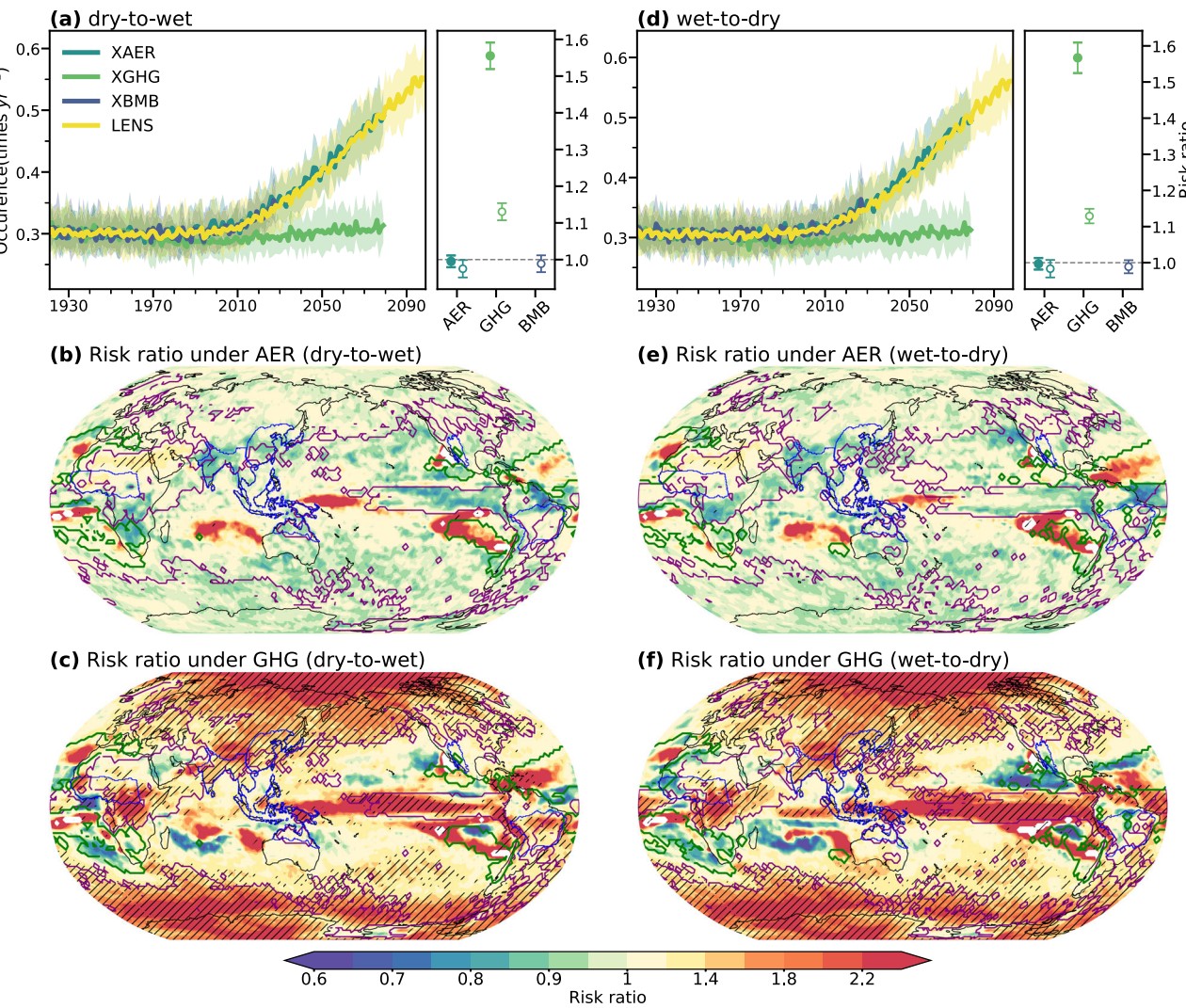

**Fig. 5 | Anthropogenic effects on changes in the occurrence frequency of precipitation whiplash.** Time series of the global area-weighted average of occurrence frequency of dry-to-wet whiplash (**a**) and wet-to-dry whiplash (**d**) derived from the ensembles of CESM-LENS (yellow), all-but-no industrial aerosols (XAERs, cyan), all-but-no greenhouse gases (XGHGs, green), and all-but-no biomass burning aerosols (XBMB, purple). The shading area shows the spread of results derived from 90% of the ensemble members. The circles at the right of each plot in Fig. 5a, d indicate the global average risk ratio of different anthropogenic forcings on the global occurrence frequency of precipitation whiplash shown in the CESM-LENS by 2028 (hollow circles) and 2079 (solid circles), and the length of the stick indicates the inter-member standard deviation. Maps are the mean risk ratio under AER (**b, e**) and GHG (**c, f**) on changes in the occurrence frequency of dry-to-wet whiplash (**b, c**) and wet-to-dry whiplash (**e, f**) over 2040–2079. Hatching shows that more than 90% of ensemble members agree on the sign of risk ratio. The green contour indicates the region where the signal-to-noise ratio (S/N) <−1, while the purple contour indicates the region where S/N >1. The blue lines indicate the monsoon regions.

110 years results in a drier climate in the regions of EAsiaM and SAsiaM, and thus 3–10% decreases in whiplash risk in these regions (Supplementary Fig. 16). AER-forced cooling compensated for the GHG-driven global warming of the 20th Century[38] and thus helped to decrease the frequency of precipitation whiplash events. However, AER forcing is expected to show a negligible impact on precipitation whiplash over most regions after the 2020s due to the expected reduction of the increase in AER emissions from the most recent period, as shown by XAER simulations on which AER emission reduces only 3% of the risk of whiplash occurrences by 2028 and even less by 2079 (Fig. 5a, d). In the future period (2040–2079), on the global land, AER is still projected to reduce the risk of whiplash occurrence in southeastern Asia, eastern and southern Africa, and northeastern South America (Fig. 5b, e). BMB shows fewer impacts on whiplash in most areas over the recent 110 years, but contributes to a 3–5% reduction in whiplash in SAsiaM, SamerM (wet-to-dry only), and AusMCM (Supplementary Fig. 16). Because of the reduced emissions[39] of aerosols and their much shorter lifetimes in the atmosphere than that of GHGs, future projection

scenarios assume that GHG warming will substantially exceed the aerosol cooling effect if GHG emissions cannot be substantially reduced[40]. The GHG-driven component will thus be the dominant anthropogenic factor causing more dramatic fluctuations in precipitation in the future.

## Large-scale circulation background during precipitation whiplash

To preliminarily investigate the physical mechanisms underlying changes in the occurrence of precipitation whiplash, we further explore the composite large-scale circulation background during the evolution of precipitation whiplash events, taking the northeastern China region (NEC) as an example. NEC is one of the regions where precipitation whiplash occurs and increases most frequently in continental Asia (Fig. 1a). The dry-to-wet (wet-to-dry) events in NEC occurred on average in early August (late June), indicating that summer (local rainy season) frequent with large precipitation anomalies in NEC (Fig. 1g, h). During the dry state in the dry-to-wet event (Fig. 6a),

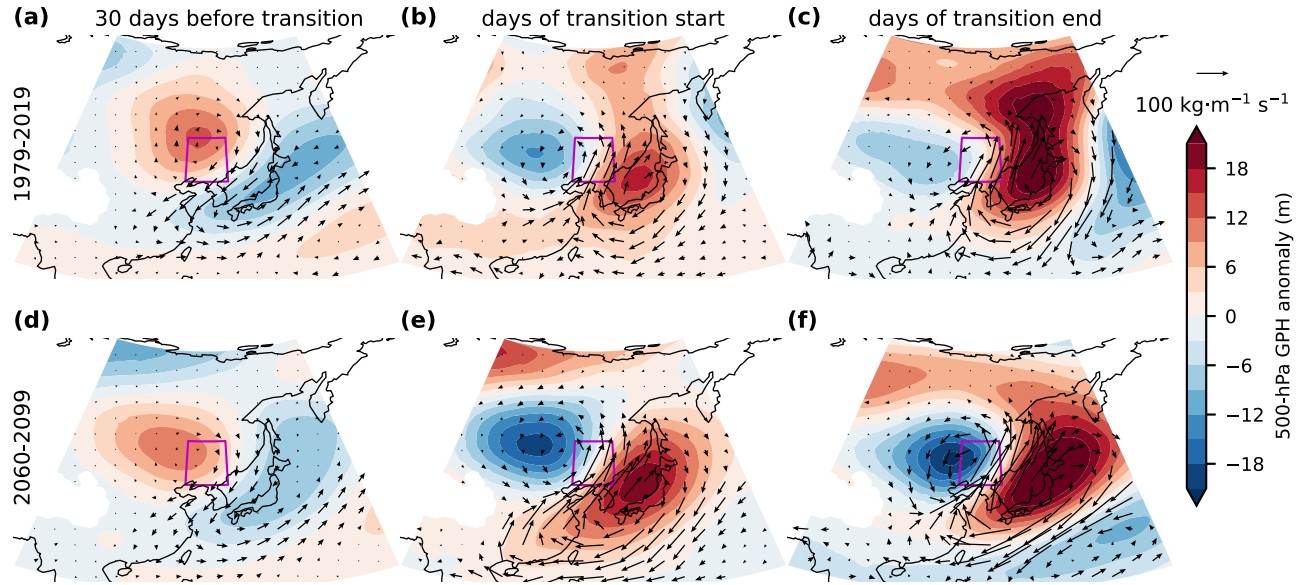

**Fig. 6 | Large-scale atmospheric circulation associated with dry-to-wet whiplash over northeastern China.** Ensemble mean atmospheric anomalies for **a**, **d** 30 days before the occurrence of dry-to-wet whiplash (controlled by dry condition), **b**, **e** the days when the transition from dry-to-wet extremes starts, and **c**, **f** the days when the transition ends (the first days controlled by wet condition) over the current period (1979–2019; **a**–**c**) and the future period (2060–2099; **d**–**f**). Composite 500 hPa geopotential height (GPH) anomalies are shown by colorful shadings, and the magnitude and direction of the column-integrated water vapor anomalies are shown by scaled arrows.

the NEC region is controlled by an anomalously strong anticyclone concurrent with a low-pressure anomaly over the Korean Peninsula and Japan and a western Pacific subtropical high (WPSH) at the low-latitudes, presenting a meridional tripolar pattern. This meridional Rossby wave train may be related to the Pacific-Japan (or East Asia-Pacific teleconnection) pattern[41]. The north-easterly wind anomaly in the northwestern part of the low-pressure center carries substantial water vapor to east-central China, while preventing the northward transport of East Asian summer monsoon (EASM) water vapor, which in turn reduces the precipitation in the NEC region. In this dry stage, the sinking airflow and enhanced atmospheric stability in the NEC region controlled by high pressure allow dry conditions to persist. Later, the high-pressure anomaly located at the low latitudes extends northward (Supplementary Fig. 20) until it is located over Japan on the days we defined as the start of the transition to the wet state, and the high-pressure anomaly previously controlling the NEC region moves to the high-latitudes, at which time the low-pressure center begins to move toward the NEC region (Fig. 6b). On the days of transition end (Fig. 6c), the NEC region undergoes the transition from dry-to-wet anomaly and is controlled by a low-pressure center. The water vapor from the outer edge of the anomalous high pressure over Japan is enhanced and transported to the NEC region, bringing continuous and long-lasting precipitation (Fig. 6c).

During the wet state of the wet-to-dry event (Supplementary Fig. 21a), an anomalously low pressure exists in the NEC region, while a high-pressure anomaly exists in its eastern part from over Japan to the Sea of Okhotsk and its northern high latitudes, transporting water vapor from the northwest Pacific Ocean to the northeast along the outer edge of the high-pressure center. However, as the southern low-pressure anomaly keeps moving northward (Supplementary Fig. 22a), the moisture transported to the NEC region from the northwest Pacific Ocean is suppressed, and more is sent to southeastern China, and the NEC region begins to be controlled by anomalous high-pressure when the transition starts (Supplementary Fig. 21b). During the 30 days after the transition days, the low-pressure center keeps advancing northward, but the intensity of the high-pressure center weakens slightly (Supplementary Fig. 22a).

The summer background climate of average water vapor transport in the NEC region is mainly driven by the EASM and the westerly jet stream[42–44]. Previous studies suggested that variations in the western edge of the WPSH may significantly affect the lower-level moisture transport in eastern China on a sub-seasonal scale, and thus influence the spatial and temporal characteristics of persistent heavy precipitation[45,46]. The EASM transports water vapor to the NEC region along the west of WPSH, so the moisture transport anomalies caused by factors such as the structure, strength, and intra-seasonal movement of the WPSH[42,47] and the time of onset, duration, and intensity of the EASM are the keys to the dry and wet anomalies in the NEC region[43,48].

In the future scenario (2060–2099; Fig. 6d–f and Supplementary Figs. 21d–f), the transitions of large-scale circulation patterns associated with dry and wet extremes are similar to the current period, suggesting that the atmospheric circulation patterns dynamically drive the occurrence of precipitation whiplash in the NEC. The positive and negative circulation anomalies are stronger, implying a potentially stronger intensity and the wider impact of precipitation whiplash. These more drastic and rapid swings between the dry and wet extremes could also be related to the enhanced seasonal hydrological cycle[14] due to an increased water-holding capacity of the atmosphere in a warming climate[49]. In addition, exploring the change in influencing factors moderating the circulation anomalies in future scenarios, including sub-seasonal variability of WPSH and its possible influencing mechanisms (e.g., SSTs[50]), the amplitude of mid-latitude Rossby wave trains (East Asia-Pacific teleconnection pattern in meridional direction[41] and Silk-Road pattern in latitudinal direction[51]), large-scale teleconnection activities[52,53], and local positive feedback processes such as cloud properties and surface energy[54] may be capable of explaining the future variability of precipitation extremes and whiplash in the NEC region, which requires further work.

## Discussion
Using various precipitation datasets obtained from observations, reanalyses, a large multi-model ensemble (CMIP6), and a large single-model ensemble (CESM-LENS), we characterize the global occurrence

frequency, transition duration, and intensity of precipitation whiplash over the historical (1979–2019) period on the regional and global scales. We also analyze projected future changes in whiplash under the scenario of the RCP8.5 emissions using simulations and assess the influence of individual anthropogenic forcings using CESM-XLENS. Collectively, both observations (six observed datasets) and simulations (CESM-LENS and CMIP6) show increases in the global (land) mean frequency of whiplash events since the late 1990s. By the end of the 21st Century, the projected global (land) mean frequency of whiplash will be $2.56 \pm 0.16$ $(3.43 \pm 0.22)$ times compared to the current period, accompanied by increasingly rapid and intense transitions between wet and dry extremes. Anthropogenic signals of the global mean (land mean) dry-to-wet and wet-to-dry whiplash features emerge in 2028 (2017) and 2033 (2017), respectively.

The polar and monsoon regions are projected to experience the most dramatic increases in whiplash frequency, with a more than 196% increase in monsoon regions located on the west of the Pacific. Changes in precipitation whiplash occurrences are concurrent with changes in precipitation totals, despite modest spatial heterogeneity of covarying trends in precipitation totals and whiplash. Percentage increases in precipitation whiplash occurrences in 2/3 of regions showing increased precipitation totals are projected to be more than 6 times than that in precipitation totals, indicating that the rate of change in precipitation whiplash is much greater than precipitation totals when the globe is becoming wetter and more variable simultaneously[13]. Our findings suggest that GHG emissions will bring about a substantial increase in the risk of precipitation whiplash occurrences (55 ± 4% up to 2079), with a drastic response in monsoon regions over the west of the Pacific (>50%) and polar regions (>120%), despite opposing influences of AER emissions on decreasing occurrences, given GHG emissions are projected to sharply outpace AER emissions after the 2020 s.

To further understand the mechanisms underlying changes in precipitation whiplash, we link the patterns and transition processes of circulation anomalies to dry and wet extremes during whiplash events. We explore the variability of large-scale processes during precipitation whiplash in the six monsoon regions mentioned, despite only showing detailed results for the NEC region here as an example. All dry-to-wet (wet-to-dry) whiplashes are well controlled by the movement of low (high) pressure center and some surrounding circulation anomalies. However, atmospheric circulation conditions for precipitation anomalies vary regionally over the world, and consequently, the mechanisms leading to precipitation extremes (dry or wet anomalies) and whiplash may differ among regions, which still needs further investigation. For example, occurrences of precipitation whiplash over NEC are often connected to the configuration and shifts of WPSH, EASM and the westerly jet stream associated with anomalously low and high precipitation. Seasonal wet conditions in California are associated with strong low-pressure anomalies in the northeast Pacific and enhanced storm tracks[55], while dry conditions are linked to persistent high pressure in the eastern Pacific[56]. In the Indian subcontinent core monsoon region, wet spell moisture is mainly derived from monsoonal depressions and cyclonic storms that often form in the Bay of Bengal and move northwestward into the monsoon trough[57]. The frequency of wet and dry events is influenced by the northward propagation of convective instability from the equatorial Indian Ocean and northwestern tropical Pacific Ocean. Therefore, the dynamic processes of the spatiotemporal distribution of precipitation whiplash in different regions cannot be summarized and require the following analysis in the context of the large-scale local circulation affecting specific regions.

In a future warming atmosphere, the tropospheric water vapor increases approximately following the Clausius-Clapeyron relationship[49,58,59], which influences the spatial pattern of changes in precipitation regimes in both thermodynamic (moisture) and dynamic (circulation) ways[11,49,60]. Thermodynamically, increased atmospheric water vapor resulting from atmospheric warming promotes negative (positive) anomalies in moisture convergence in climatologically dry (wet) regions where downward (upward) motion is predominant, creating unfavorable (favorable) conditions for processes that trigger precipitation. Dynamically, a change in precipitation alters the release of latent heat, and thus the corresponding vertical motion, and further alters precipitation through this dynamic effect associated with changes in circulation[11,49]. A pattern of increasing wetting in wet areas and drying in dry areas is found in both observations and simulations[61–65]. The warming effect not only leads to changes in the mean[58,66] and extreme[67,68] precipitation characterized by strong spatial heterogeneity, but also has a significant influence on precipitation variability[7,12]. For example, the seasonal variability in global monsoon precipitation increases with global warming and is mainly related to increased moisture convergence and surface evaporation[69]. In subtropical North America, the increase in sub-seasonal precipitation variability due to increased background moisture is offset by the effect of weakening circulation variability[15], which is consistent with our projection that there is no significant change in whiplash events in the North American monsoon region. Therefore, projected changes in the characteristics of precipitation whiplash events are inseparable from changes in atmospheric moisture and circulation in a warming future climate context. Quantifying how thermodynamic and dynamical changes in a warmer climate exacerbate or weaken the whiplash between regional sub-seasonal-scale dry and wet extremes is practical for a comprehensive analysis to understand the underlying mechanisms of precipitation whiplash.

Although humans are adapting to regional regimes of scarce or excess precipitation, this adaptation will be complicated by ongoing changes in precipitation regimes. The more frequent, intense, and rapid dry-wet transitions resulting from the increasing temporal variability of sub-seasonal-scale precipitation will further challenge water resource management and disaster prevention for human society. Natural ecosystems could also be considerably impacted by this novel regional precipitation regime. This study is primarily concerned with the superposition of sub-seasonally to seasonally persistent and widespread dry or wet extremes that have resulted in a double whammy to agricultural yields, water quality, and safety of life and property. A potential limitation is that our method of using sliding windows to accumulate sub-seasonal precipitation may smooth out short-lived, intense rainstorm events that can cause flooding, transportation disruptions, damage to urban infrastructure, and loss of life[70]. Events such as short-duration intense rainstorms following a prolonged drought that poses a significant threat to the emergency response system can be explored at more flexible spatiotemporal scales in subsequent studies. Recognizing the all-around risk induced by more volatile precipitation and more whiplash under multiple timescales is the first step to developing a resilient coupled natural and human system in a warming climate.

## Methods
### CESM-LENS simulation
We use the National Center for Atmospheric Research (NCAR) Community Earth System Model Large Ensemble (CESM-LENS), an ensemble of fully coupled global climate model simulations, which provides 40 experiments forced with all historical radiative (1920–2005) and the RCP8.5 scenarios (2006–2100). The CESM-LENS[33] is a large ensemble GCM simulation for the study of internal climate variability and forecast uncertainty[71] on long timescales. Each member is generated by randomly perturbing temperatures at the level of round-off error while keeping the same radiative forcing scenario (historical up to 2005 and Representative Concentration Pathway (RCP8.5) thereafter)[72,73]. Therefore, compared to historical datasets that are restricted to shorter series and other smaller ensembles, CESM-LENS

provides a sufficiently long daily precipitation dataset ensemble of high resolution (~1°) for analyses of the internal variability of precipitation. CESM-LENS allows us to directly assess changes in precipitation extremes we are interested in both historical and future periods, and enables us to examine the robustness of changes in precipitation extremes across a wide range of simulated internal climate variability. We used daily precipitation output from 40 members of the twentieth century (20 C; 1920–2005) and representative concentration pathway 8.5 (RCP8.5; 2005–2100) climate scenario[74]. Precipitation outputs for each ensemble member are firstly re-gridded to a uniform 2° grid before calculating the extreme precipitation indices (see the following on Identification of Precipitation Whiplash). The ensemble mean precipitation whiplash characteristics were adopted to show the projected forced changes, and differences between member responses represent the internal climate variability of the results.

## CESM-XLENS simulation

To further examine specific human activities on changes in global and regional precipitation, we use precipitation data derived from the CESM large ensemble (LENS) "single forcing" experiments (CESM-XLENS)[72], which is recently extensively adopted for attributing extreme climate change to an individual anthropogenic forcing such as industrial aerosols (AER), greenhouse gages (GHG), and aerosols from biomass burning in agriculture and wildfires (BMB)[38,75–77]. The CESM-XLENS simulations were conducted using the same configuration and initialization protocols as the original CESM-LENS experiment[33], except keeping the AER, GHG, and BMB conditions fixed at the level of 1920, respectively[34], while natural forcing factors (i.e., solar and volcanic), and all other external anthropogenic forcings (i.e., stratospheric and tropospheric ozone, land use/land cover changes and individual forcings other than the one that was fixed in 1920) follow historical and RCP8.5 scenarios as the original CESM-LENS experiment did[33]. The XAER (fixed industrial aerosols, 1920–2080), XGHG (fixed greenhouse gas, 1920–2080), and XBMB (fixed aerosols from biomass burning in agriculture and wildfires, 1920–2029) forcing simulations we used have 20, 20, and 15 ensemble members, respectively. Precipitation outputs for each ensemble member are firstly re-gridded to a uniform 2° grid as to the CESM-LENS ensembles. We analyzed precipitation whiplash events for each ensemble member of CESM-XLENS.

## CMIP6 ensemble

We also used precipitation output from climate model simulations generated as part of the Coupled Model Intercomparison Project Phase 6 (CMIP6) project[35] for comparison with CESM-LENS. We used a multi-model ensemble consisting of 55 realizations of 22 distinct climate models (See Supplementary Table 1) from the CMIP6 project over the same period (i.e., 1920–2014 for the historical period and 2015–2100 for the SSP5-8.5 emission scenarios whose expected radiative forcing level in 2100 is 8.5 W m$^{-2}$ similar to RCP8.5), and use their precipitation outputs to compare with the CESM-LENS simulations. Different from CESM-LENS, whose ensemble model uncertainty consists of internal climate variability only, CMIP6 composes a combination of internal climate variability and model formulation differences (i.e., structural uncertainty) with unclear relative importance[78,79]. We first compare the model spread of CESM-LENS and CMIP6 (Text S1 in supplementary materials), and the results show that the spread of the CMIP6 ensemble is representative of the internal variability generated by CESM-LENS in our study (Supplementary Text 1 and Supplementary Fig. 23). The CMIP6 subset we selected is suitable for use with CESM-LENS to enhance the robustness of the results. Precipitation outputs for each model member are firstly re-gridded to a uniform 2° grid and then precipitation whiplash are identified. The ensemble means of each respective model are calculated weight equally after identifying extreme events.

## Observations

To compare with the climate model simulated events, we identify observed whiplash events using daily precipitation outputs from three reanalysis, one satellite-based dataset and two ground-based datasets. Reanalysis are the fifth generation of the European Centre for Medium-Range Weather Forecasts (ECMWF) Reanalysis (ERA-5[80]), the Modern-Era Retrospective Analysis for Research and Applications, Version 2 (MERRA-2[81]), and the Japanese 55-year Reanalysis (JRA-55[82]). The satellite-based dataset is the Climate Hazards Group Infrared Precipitation with Stations (CHIRPS[83]). Ground-based datasets are Global Precipitation Climatology Centre Full Data Daily Product Version 2022 (GPCC[84]), and Rainfall Estimates on a Gridded Network (REGEN[85]). A summary of these gridded precipitation datasets used in this study is shown in Supplementary Table 2. The above raw precipitation data are also re-gridded to a uniform 2° grid. Hereinafter, the results based on the above datasets are referred to as precipitation "observations".

## Identification of precipitation whiplash

Since regions showing increases (decreases) in precipitation in the long-term period will substantially magnify the absolute exceedance (deficit) of precipitation totals in the future, raw precipitation data is first detrended (Supplementary Fig. 1). The cumulative precipitation totals are calculated by the 30-day rolling sum of daily precipitation throughout the entire period. To standardize the data and eliminate the effects of precipitation seasonality, the annual-cycle of the time series of precipitation totals at the sub-seasonal scale is removed (Supplementary Fig. 2). In this study, to make the characteristics of precipitation regimes comparable between various ensembles and forcings, we adopt low and high threshold values obtained from the 10th and 90th percentile of standardized precipitation anomalies over the current period (1979–2019) for dry and wet extremes, respectively. Finally, a transition from the lower (upper) threshold to the upper (lower) threshold is defined as precipitation whiplash of dry-to-wet (wet-to-dry). We analyze the sensitivity of results to the number of days (20, 25, 30, 35, and 40 days) used for rolling sum calculation and different thresholds (80th, 90th, and 95th), and it shows that our conclusions are not dependent on the length of the rolling time period Supplementary Fig. 4) and extreme quantile threshold (Supplementary Fig. 5). We focus on three occurrence characteristics of precipitation whiplash events, the occurrence frequency, transition duration, and intensity. The occurrence frequency is the number of precipitation whiplash events that occurred in a period of time. The transition duration is defined as the duration of the last day of the dry (wet) extreme to the first day of the wet (dry) extreme in a whiplash event. The transition intensity is defined as the absolute value of the difference between anomaly values of the driest and the wettest days during a whiplash event (Supplementary Fig. 2c).

We splice some dry or wet extreme events that are briefly interrupted and then continue immediately according to a method similar to run theory[82]. For example, if two dry (wet) extreme events are interrupted by a short interval, we splice the two briefly interrupted events together if the average of the first dry (wet) extreme period and the interrupted period is still lower (higher) than the 10th (90th) extreme threshold. We remove dry or wet extreme events that last no more than three days to ensure that the dry and wet extremes of interest are not transitory. We only consider whiplash events that shift from dry-to-wet or wet-to-dry within 30 days, because the cumulative precipitation of two opposite events beyond this time length does not overlap temporally and, therefore cannot be considered a rapid transition.

## Detrending

We detrend the original precipitation time series to remove the effects of long-term climate change for each grid cell. Taking a grid cell in CESM-LENS as an example, we use a linear fit to the annual mean

precipitation of the historical radiative (1920–2005) and the RCP8.5 scenarios (2006–2100) data for each grid cell, and then separate the short-term precipitation variability from the long-term climate change by subtracting its corresponding fitted trend value from the daily precipitation data for each year. We compare three different detrending methods, including the simple linear detrending of two series, polynomial (binomial) fit detrending of the entire time series (historical + RCP8.5), and a "scaling" method that estimates the trend based on regression of model ensemble mean and individual ensemble members. The "scaling" method assumes that the forced response can be estimated by averaging over model members[78]. The precipitation data (Supplementary Fig. 1c) and whiplashes (Supplementary Fig. 1d, e) using different detrending methods present very similar results (Supplementary Fig. 3). To be able to apply to the "observation" dataset, we choose the linear detrending method.

### Annual-cycle-removing

Cumulative precipitation totals (precipitation at sub-seasonal scale) are then calculated from a 30-day rolling sum of detrended daily precipitation over the entire period (black line in Supplementary Fig. 2a). Standardized annual-cycle-removed cumulative precipitation anomalies (black line in Supplementary Fig. 2b) are calculated by Eq. (1).

$$P_{ij'} = \frac{P_{ij} - \bar{P_j}}{\sigma_j} \tag{1}$$

where $P_{ij}$ is the cumulative precipitation totals of the $j^{th}$ ($j = 1, ..., 365$) Julian day in year $i$ ($i = 1920, ..., 2100$), $\bar{P_j}$ is the 181-year (1920–2100) mean of the $j^{th}$ Julian day, and $\sigma_j$ is the 181-year standard deviation of the $j^{th}$ Julian day. After the above processing, the detrended and annual-cycle-removed precipitation totals at the sub-seasonal scale for each grid lower (higher) than the lower (upper) threshold values are defined as dry (wet) extreme events (Supplementary Fig. 2b, c).

### Climatology of the occurrence frequency and timing of precipitation whiplash

To obtain the climatology of precipitation whiplash (Fig. 1 and Supplementary Figs. 6, 7), we derive the means of the occurrence frequency and timing in the CESM-LENS, CMIP6, six gridded precipitation datasets over the current period, i.e., 1979–2019, to investigate their global climatology. We define the average timing of the events by the average date on which events have occurred. We calculate the average day within a year on which whiplash has occurred during the period of interest[86] for each grid. We first convert the date of occurrence $D_i$ of a whiplash event in year $i$ into an angular value $\theta_i$ by

$$\theta_i = D_i \cdot \frac{2\pi}{m_i} \quad 0 \le \theta_i \le 2\pi \tag{2}$$

where $D_i = 1$ corresponds to January 1 and $D_i = m_i$ corresponds to December 31, and where $m_i$ is the number of days in that year. The average date of occurrence $D$ of whiplash at a grid is then defined as:

$$\bar{D} = \begin{cases} \tan^{-1}\left(\frac{\bar{y}}{\bar{x}}\right) \cdot \frac{\bar{m}}{2\pi} & \bar{x} > 0, \bar{y} \ge 0 \\ \left[\tan^{-1}\left(\frac{\bar{y}}{\bar{x}}\right) + \pi\right] \cdot \frac{\bar{m}}{2\pi} & \bar{x} \le 0 \\ \left[\tan^{-1}\left(\frac{\bar{y}}{\bar{x}}\right) + 2\pi\right] \cdot \frac{\bar{m}}{2\pi} & \bar{x} > 0, \bar{y} < 0, \end{cases} \tag{3}$$

with

$$\bar{x} = \frac{1}{n}\sum_{i=1}^{n} \cos(\theta_i) \tag{4}$$

$$\bar{y} = \frac{1}{n}\sum_{i=1}^{n} \sin(\theta_i) \tag{5}$$

$$\bar{m} = \frac{1}{n}\sum_{i=1}^{n} m_i \tag{6}$$

where $x$ ($y$) is the cosine (sine) components of the average date, and $n$ is the total amount of whiplash at that grid.

### Future changes in the occurrence characteristics of precipitation whiplash

We calculate the relative changes in the occurrence characteristics of extreme events ($F$) in a given year $y$ relative to the current period (1979–2019) by Eq. (7) (Fig. 2b, d, f and Supplementary Figs. 9–11).

$$\Delta F_y = \frac{F_y - F_{current}}{F_{current}} \times 100\% \tag{7}$$

Similarly, the relative changes in the occurrence characteristics of extreme events within two given periods (*P1* for the earlier one and *P2* for the later one) can be calculated by Eq. (8) (Fig. 2a, c, e and Supplementary Figs. 8–10; Figs. 3, 4 and Supplementary Figs. 14, 15).

$$\Delta F_{P2} = \frac{F_{P2} - F_{P1}}{F_{P1}} \times 100\% \tag{8}$$

### Signal-to-noise ratio

We calculate the signal-to-noise ratio[87,88] (S/N hereafter) to assess the relative contribution of the internal climate variability and the forced response (Figs. 3, 5 and Supplementary Figs. 12–15), as

$$S/N = \frac{\overline{\Delta F}}{\sigma \Delta F} \tag{9}$$

where $\Delta F$ is the forced response which is the ensemble mean frequency of whiplash obtained from 40 (55) members of CESM-LENS (CMIP6). The noise is defined as $\sigma \Delta F$, the inter-member standard deviation of the projected change in frequency of precipitation whiplash for a given time horizon. In a given radiative forcing scenario, the noise in CESM-LENS represents uncertainties that arise solely from internal climate variability, whereas in CMIP6, the noise stems from a combined uncertainties of internal climate variability and model formulation differences. The absolute value of *S/N* greater than 1 (less than 1) implies that the effect of external forcing is stronger (weaker) than that of uncertainties, i.e., the effect of external forcing does (does not) emerge.

### Random resampling method for obtaining the minimum number of members required

Based on the CESM-LENS and CMIP6 simulations, we calculate the number of ensemble members needed to obtain a detectable signal[87] (the absolute value of S/N greater than 1). For each $n$ ranging from 2–39 for CESM-LENS and 2–54 for CMIP6, we generate 100 000 new ensembles of randomly selected $n$ members. Next, we retain $n$ when 95% of the 100 000 ensembles yielded the absolute value of S/N greater than 1. The minimum number of ensemble members needed to obtain a robust precipitation whiplash change over the time series is defined as the minimum value of $n$.

### Anthropogenic influence on precipitation whiplash

We calculate the risk ratio[38] (RR), for each member, $i$, in each ensemble of CESM-XLENS, *X*, to quantify the impact of each anthropogenic

forcing on the risk of extreme whiplash characteristics (Fig. 5 and Supplementary Figs. 16–19), as

$$RR = \frac{F_{LENS}}{F_X^i} \qquad (10)$$

where $F_X^i$ denotes the whiplash characteristics in member $i$ of the ensemble with forcing $X$ fixed in 1920 (e.g., XAER) and $F_{LENS}$ indicates the ensemble mean of CESM-LENS.

The contribution of each anthropogenic forcings $X$ on the change of occurrence characteristics of precipitation whiplash is estimated by Eq. (11).

$$IF_X = \frac{C_{LENS} - C_X}{C_{LENS}} \times 100\% \qquad (11)$$

where $C$ denotes the relative change of the whiplash characteristics in the future period (2040–2079) relative to the current period (1979–2019).

### Large-scale atmospheric circulation of precipitation whiplash
We analyze composite maps of geopotential heights (GPH) anomaly at the 500 hPa level and vertically integrated vapor transport anomaly for precipitation whiplash using the 40 CESM-LENS ensembles (Fig. 6 and Supplementary Figs. 20–22). We compare the atmospheric anomaly before the occurrence of dry-to-wet (wet-to-dry) whiplash, i.e., the days controlled by dry (wet) condition, and at the transition day, and after the occurrence, i.e., the days controlled by wet (dry) condition. We examine the relative robustness of the atmospheric conditions leading up to the whiplash by comparing the differences in the atmospheric circulation anomaly between the current period (1979–2019) and the future period (RCP8.5 simulation; 2060–2099). The anomalies are calculated by subtracting the climatological average for the time period.

Vertically integrated vapor transport (IVT) is defined as:

$$IVT = \frac{1}{g} \int_{p_s}^{0} uq dp \qquad (12)$$

where $u$ is horizontal (zonal or meridional wind), $q$ is specific humidity in a given vertical pressure level, and $p$ is the pressure value. $p_s$ is surface pressure and $g$ is the gravitational acceleration (9.806 m s$^{-2}$).

### Monsoon regions
This study defines six monsoon regions with reference to the sixth Assessment Report (AR6) of Intergovernmental Panel on Climate Change (IPCC), namely North American monsoon (NAmerM), South American monsoon (SAmerM), West African monsoon (WAfriM), South and Southeast Asian monsoon (SAsiaM), East Asian monsoon (EAsiaM) and Australian-Maritime Continent monsoon (AusMCM) regions. The results for the globe and different monsoon regions are area-weighted averages.

## Data availability
CESM-LENS data are made available by the CESM Large Ensemble Community Project (https://www.cesm.ucar.edu/projects/community-projects/LENS/data-sets.html) and CESM-SF data are made available by the CESM1 "Single Forcing" Large Ensemble Project (https://www.cesm.ucar.edu/working_groups/CVC/simulations/cesm1-single_forcing_le.html). The CMIP6 ensemble used for this study are freely available from the Earth System Grid Federation (ESGF, https://esgf-node.llnl.gov/search/cmip6/). ERA-5 data were obtained from https://cds.climate.copernicus.eu/, JRA-55 data were obtained from https://jra.kishou.go.jp/JRA-55/index_en.html, MERRA-2 data were obtained from http://gmao.gsfc.nasa.gov/reanalysis/MERRA-2, CHIRPS data were obtained from https://www.chc.ucsb.edu/data/chirps, GPCC data were obtained from https://climatedataguide.ucar.edu/climate-data/gpcc-global-precipitation-climatology-centre, and REGEN_LongTermStns data were obtained from https://geonetwork.nci.org.au/geonetwork/srv/eng/catalog.search#/metadata/f6973_9398_8796_3040. Maps have been made with vector files from https://www.naturalearthdata.com/. The datasets generated in this study have been deposited in the following Zenodo repository[89]: https://doi.org/10.5281/zenodo.7653038. This repository also includes all the source data necessary to reproduce all the figures in this study.

## Code availability
The code used in this study, including the code to reproduce all the figures in this study, can be found in the following Zenodo repository[90]: https://doi.org/10.5281/zenodo.7813096.

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

## Acknowledgements

This work was supported by the National Natural Science Foundation of China (52179030 and 51809295, X.T.), the National Key R&D Program of China (2021YFC3001000, X.T.), and the Guangzhou Science and Technology Plan Project (Grant No. 201904010097, X.T.). We are very grateful to ComputeCanada for providing the storage and computing resources for this analysis.

## Author contributions

X.T. conceived the study. X.T. and X.W. designed the study, performed the analyses, interpreted the results, and wrote the paper. Z.H., J.F., X.T., T.Y.G., S.D., Y.L., and B.L. processed the data, interpreted the results, and wrote the paper.

## Competing interests

The authors declare no competing interests.
