## [Peer Review File · Nature Communications]

Increasing global precipitation whiplash due to anthropogenic greenhouse gas emissionsREVIEWER COMMENTS

Reviewer #1 (Remarks to the Author):

Using CESM1 all forcing and single forcing large ensembles, CMIP6 multimodel ensemble simulations, and multiple reanalysis datasets, the authors conduct a global assessment of sub-seasonal precipitation whiplash (dry-to-wet or wet-to-dry transition of extreme events). The authors show the climatology and the historical changes of the whiplash events. Mainly focusing on monsoon regions, the authors assess the change in whiplash occurrence due to all radiative forcings and individual anthropogenic radiative forcings. Finally, the mechanisms behind the whiplash events over northeastern China are investigated.

The global analysis of the historical and projected future changes of sub-seasonal whiplash events, and the assessment of the influence of individual radiative forcings are novel. The results presented are of high societal relevance. Overall, the manuscript is well organized without any significant flaws in the logic; the figures are clearly presented, and the results support the conclusions. I only have a few minor comments for the authors.

Line-by-line comments:

L6-8: "detect and attribute" implies that this is a detection and attribution study, which typically detects the signal in observations using earth system model simulations. As this study only assesses the influence from individual radiative forcings in CESM1 simulations, not in observations, I suggest rewording here to reflect that.

L21: How about dried instead of "baked"?

L73: consistent instead of "consistently"?

L87: should "in the Northern Hemisphere" be omitted?

L125: Describing the regions in the figure 3 caption or in the main text (or at least refer to methods) would be helpful for the readers.

L139: illustrates instead of "illustrating"?

L152: How about the margin between the reduced and increased precipitation ..., instead of "the junction ..."?

L165: is it wet-to-dry or dry-to-wet here? Based on the previous sentence and Fig. S13, I believe it should be the latter.

L166: The section "For the other four monsoon regions," makes this sentence confusing. Consider omitting or rewording.

L171: has been increasing, instead of "have been increasing"?

L189: Refer to Fig. S12 here.

L198: Arrows in Fig. 6 and Fig S14 are barely visible. Increasing their size would be helpful.

L198-202: The sentence seems too long. How about removing "and" in L201 and starting a new sentence?

L233: Reference 44 is incomplete (L551).

L237: Please consider removing ", etc.,".

L240-245: As the first comments, I recommend rewriting this part to reflect the fact that the

authors assessed the influence of individual forcings in CESM1 simulations, not in observations/reanalyses.

L249-253: I suggest moving this section to the end of the discussion section (at L271).

L340-341: Do you think the results would be different if you first remove the seasonal cycle and the trend from precipitation data, prior to calculating the whiplash events?

L348: Should it be "lower and higher" instead of "lower (higher)"?

L351: Should wet-to-dry and dry-to-wet be swapped?

L383: is the part "(10th or 5th)" necessary?

Reviewer #2 (Remarks to the Author):

General comments:

In my opinion, this manuscript is well written and organized with good quality figures to present several important findings: i) precipitation whiplash in the current climate and projected future climate, ii) concurrent changes in precipitation total and whiplash, iii) attribution of precipitation whiplash changes to different anthropogenic forcings, and iv) large-scale atmospheric condition for precipitation whiplash in the current and projected future climate. Despite the vast results with multiple model simulation sets, the results are concisely and effectively presented. I think this manuscript deserves to be published in Nature Communications after minor revisions. The specific suggestions for improving the manuscript are as follows.

Minor revisions recommended:

1. This manuscript uses reanalysis precipitation products (ERA5, JRA55, and MERRA2) as observational precipitation data. I think at least one satellite-based precipitation product should be added to the analysis because reanalysis precipitation products include errors from model simulations.
2. [L24] It would be good to use the word "nitrogen" instead of "N" to avoid ambiguity.
3. [L47] Remove a typo "."
4. [L87-89] It seems that this sentence should be revised because there is redundant information for regions.
5. [L111] "Fig. 2c-d" might be "Fig. 2a-b".
6. [L139] "illustrating" might be "illustrates".
7. [L148] Which land region do you mention? Do you mean the Arctic land region? Please clarify it.
8. [L165] "wet-to-dry" might be "dry-to-wet".
9. [L200] "western subtropic Pacific high (WPSH)" might be "western Pacific subtropical high (WPSH)".
10. [L224-225] It would be great if the authors could explain a little bit more about why the circulation anomalies will be deeper and more extensive in the future scenario. I think it would be an important step forward for better understanding the underlying mechanisms in changes in precipitation whiplash.
11. [L251] Could you explain a little bit more what novel regional precipitation regimes are?
12. [Discussion and Conclusions section] The authors discussed atmospheric circulation conditions for extreme events in the current climate. However, one of the main objectives of this study is to investigate future changes in extreme events. So, I think the author should also discuss the future changes in atmospheric circulation conditions for extreme events.
13. [L305] Could you describe a little bit more what natural forcing factors are and what other external anthropogenic factors are?
14. [L323-324] In the context, precipitation extremes are firstly identified in native resolution, and then the results were re-gridded to a uniform 2-degree resolution. However, many previous studies perform re-gridding at the very first and then proceed following analyses. Do you have any specific reasons to do re-gridding after identifying precipitation extremes? Have you compared the results from those different processes?

15. [L348] Please explain a little bit more how you processed detrend and annual-cycle-removing at the subseasonal scale and how you defined the subseasonal scale.
16. [L349] "lower and upper" might be "lower (upper)"
17. [L361-367] This might not fit in the method section.
18. [L360-361] I think more explanation is needed here. How did you average dates (1/1, 1/2, ... 12/31)?
19. [L373, L380] "Fig. 2c-f" might be "Fig. 2c-d".
20. [L398] In my understanding, F_i is an experiment without forcing i . If so, the text should be revised to clarify it.
21. [L1 in Fig.1 caption] "precipitation whiplash" might be "precipitation extremes and whiplash".
22. [Fig. 2 caption] The information in parentheses in L3-4 and L5-7 are redundant.
23. [L8 in Fig.3 caption] "CESM-LENS ensemble" might be "CESM-LENS ensemble and CMIP6 ensemble"
24. [Fig. 4] I think reorganizing the position of color bars will help better perceive the figure. It would be great if the authors could reorganize the color bars with 2x2 rows and columns as below.
morePmoreE(top left), morePfewerE(top right)
lessPmoreE(bottom left), lessPfewerE(bottom right)
25. [Fig. 6] It is difficult to identify the directions of the arrows on the figure. I think the quality of the arrows should be improved.

Reviewer #3 (Remarks to the Author):

Review of the manuscript entitled “*Increasing global precipitation whiplash due to anthropogenic greenhouse gas emissions*” for Nature Communications

Specific recommendation: Rejection

The paper aims at detecting and attributing changes in subseasonal precipitation “whiplash” over the 1920-2100 period, based on a large ensemble of CESM simulations with all or individually-removed anthropogenic forcings and a CMIP6 multi-model ensemble (54 realizations of 22 models). Anthropogenic GHG emissions are found to dominate the projected changes, with a 20% global increase in the whiplash occurrences that outpace the effect of the anthropogenic aerosols (AER) after the 2020s. A regional analysis is also conducted over six monsoon regions. Finally, the authors claim that the subseasonal shifts in precipitation are associated with shifts in circulation patterns, as illustrated by a composite analysis over Northeast China (NEC)

The paper proposes an original study based on a large number of data sets, but it suffers from several weaknesses and does not add much value to previous studies that have already quantified an increase in rainfall variability at multiple timescales (e.g., Pendergrass et al., 2017). **I therefore recommend the study to be published in another journal or to be resubmitted to Nature Communications following some of the following suggestions.** Please, find hereafter my major comments as well as a brief (non-comprehensive) list of more specific comments.

Major comments:

1) Detection-attribution:

While the authors claim in their abstract that they “*detect and attribute changes in precipitation whiplash*”, they just assess the effect of individual anthropogenic forcings in the CESM model. As defined more than a decade ago by an IPCC guidance paper (Hegerl et al., 2010), “detection of change” is defined as the process of demonstrating that climate has changed in some defined statistical sense and, therefore, primarily relies on observations. Here, no observational dataset is used. Three global atmospheric reanalyses are used to evaluate the 1980-2014 climatology of the CESM and CMIP6 models in Fig. 1, but these are not observations and there is no attempt to perform a formal D&A analysis (reanalyses are not even shown in subsequent Fig. 2, 3, 5 and 6).

If the authors decide to resubmit this study, I would thus suggest to focus the analysis over land (given the motivations), to use gridded observations (e.g., REGEN) in addition to reanalyses, and to perform a formal D&A study (or at least to compare simulated and observed trends).

2) Whiplash definition:

While the concept of “precipitation whiplash” is original and may deserve a particular attention, the motivations are not entirely convincing and the definition raises a number of questions (so that those events may correspond to different phenomena across different regions or models). A whiplash is defined as an abrupt shift between contrasted precipitation extremes, which means that the authors first need to define precipitation extremes. Their definition is based on the exceedance (deficit) of precipitation accumulated over 30 consecutive days above (below) the 90th (10th) preindustrial percentile. A linear detrending and annual cycle removing is applied to the 30-day running mean precipitation timeseries before identifying extremes in order to eliminate the effect of precipitation trends and seasonality. The motivations highlight potential impacts of such abrupt shifts on ecosystems, erosion or water quality, but seems to ignore that the succession of two extremely wet or two extremely dry months may have even stronger impacts. The authors claim (but actually don’t show) that their results are not sensitive to the number of consecutive days (within the 20 to 40-day interval), but some of the impacts that motivate their study may occur at either shorter or longer timescales, given the asymmetry of wet versus dry extremes (short-duration heavy precipitation events can be enough to trigger large impacts).

Beyond the selected 30-day timescale (which does not add much value to using daily instead of monthly data and raises the question of selecting independent extreme events given the use of running mean daily precipitation), there are other arbitrary choices that may need further justifications or explanations. Linear detrending is not necessarily the most suitable choice to remove changes in mean precipitation given the time-varying effect of AER and the availability of multiple realizations (at least this could be checked with CESM). Changes in precipitation variability have been already documented (see for instance the ARG6 WG1 assessment from both Chapter 8 and 11) so that the focus could be more specifically on “whiplash occurrences” by computing time-varying (rather than fixed preindustrial) percentiles. So doing, the number of subseasonal wet/dry extremes would not change over time, but the shifts could still change (in terms of both occurrences and seasonality, cf. Fig.1) thereby highlighting an original result beyond the increase in variability.

Finally, the distinction of “wet to dry” versus “dry to wet” transitions is so far not fully justified given the apparent similarity between their climatologies (Fig. 1) and responses (Fig. 2), except over the WafriM region (cf. Fig. 3) which may need further explanations. This strong symmetry between the two types of precipitation whiplash may be linked to their definition and may suggest alternative definitions where wet extremes do not necessarily last as long as the selected dry extremes (here 30 days). In this case, and beyond changes in occurrences and seasonality, other changes in these events could be analysed such as the mean duration of wet extremes and the mean precipitation budget across such transitions (increasing excess or deficit?).

If the authors decide to resubmit this study, I would thus suggest to use another and potentially more fruitful definition of a subseasonal precipitation whiplash.

3) CESM versus CMIP6 ensembles

The authors make use of the agreement between two ensembles to highlight the reliability of their results (e.g., L94-96). So doing they seem to ignore the different nature of the two ensembles despite a reasonable description in the Methods section. The CESM-LENS large ensemble is aimed at quantifying the contribution of internal climate variability to projected changes in key variables, while the CMIP6 ensemble (especially the way it is used in the present study) allows the authors to document the inter-model spread. The fact that CESM is generally relatively close to the CMIP6 multi-model ensemble (MME) mean does not necessarily mean that the results are reliable, especially if the CMIP6 MME spread is substantial (it only means that CESM is fairly representative of the CMIP6 MME mean climatology or response). The stippling/hatching (hatching only could be enough) shown in Fig. 2 and 4 only shows the statistical significance of the changes projected by CESM (compared to internal variability) and the global maps should be shown for CESM and CMIP6 separately. Moreover, the captions are also not clear about the shading around the CMIP6 MME mean response in both Fig. 2 and 3. Should we conclude that internal climate variability within CESM is as strong as the inter-model spread within CMIP6?

If the authors decide to resubmit this study, they may want to pay more attention to the signal-to-noise ratio and use CESM to diagnose what is the minimum ensemble size needed to detect simulated changes in whiplash occurrences (in a high-emission scenario, a larger size would be probably needed for lower-emission scenarios). Then they could select a subset of CMIP6 models that satisfy this criterion in order to assess the intermodel spread and discuss the reliability of their results.

4) Role of shifts in large-scale circulation:

In a final subsection, the authors analyse the large-scale circulation background during precipitation whiplash, taking NEC as an example. They suggest a key role of moisture transport, which is not a big surprise. My feeling is that this section is not mature enough to be useful in the submitted manuscript. The authors recognize that this is a preliminary investigation and that the mechanisms leading to precipitation whiplash may differ among regions. They also conclude that such an analysis should be done in the context of circulation changes affecting specific regions.

If the authors decide to resubmit this study, they may need to choose between two options: 1) assess the role of large-scale circulation much more thoroughly and in the context of a specific region as a separate paper; 2) develop a more generic analysis framework, potentially applicable at the global scale, aiming at isolating the contribution of changes in large-scale circulation to changes in the occurrence frequency of precipitation whiplash. My feeling is that option 2 would be very ambitious so that option 1 may be more tractable and would allow the authors to focus more thoroughly on my previous comments before resubmitting their study.

Specific comments:

I also have a number of specific comments that I won't detail at this stage given the recommended rejection of the current manuscript. They include:

- the need to improve Figure captions;
- multiple needed edits to the text;
- missing relevant references;
- lack of confidence intervals when providing global (or regional) estimates of the projected changes;
- the ensemble size of the CESM-XLENS simulations;
- RCP8.5 vs SSP5-8.5 scenario;
- a possible analysis of changes in whiplash seasonality (given the present-day climatology shown in Fig. 1);
- a possible explanation of the lack of increase after 2060 (L102);
- a possible discussion of the limitations of the “dry-get-drier” paradigm;
- a possible explanation of the contrasted response shown in Fig. 3 over West Africa;
- etc...

Reviewer #1

Using CESM1 all forcing and single forcing large ensembles, CMIP6 multimodel ensemble simulations, and multiple reanalysis datasets, the authors conduct a global assessment of sub-seasonal precipitation whiplash (dry-to-wet or wet-to-dry transition of extreme events). The authors show the climatology and the historical changes of the whiplash events. Mainly focusing on monsoon regions, the authors assess the change in whiplash occurrence due to all radiative forcings and individual anthropogenic radiative forcings. Finally, the mechanisms behind the whiplash events over northeastern China are investigated.

The global analysis of the historical and projected future changes of sub-seasonal whiplash events, and the assessment of the influence of individual radiative forcings are novel. The results presented are of high societal relevance. Overall, the manuscript is well organized without any significant flaws in the logic; the figures are clearly presented, and the results support the conclusions. I only have a few minor comments for the authors.

We wish to sincerely thank Reviewer #1 for his/her extensive and thoughtful comments which helps to improve our manuscript. We have revised the original manuscript (OM) following suggestions of Reviewer #1. There have been textual changes in the manuscript, mostly in sections of Results, Discussion and Conclusions, and Methods. The response to each point raised is given alongside the relevant specific comment below. Throughout, *reviewer comments* are in *blue* font and *italic* type, and our point-by-point response is in black font.

Line-by-line comments:

1. L6-8: “detect and attribute” implies that this is a detection and attribution study, which typically detects the signal in observations using earth system model simulations. As this study only assesses the influence from individual radiative forcings in CESM1 simulations, not in observations, I suggest rewording here to reflect that.

Based on the reviewers' suggestions, we have changed the wording here to:

Lines 5-7: We detect historical and projected future changes in whiplash and investigate the role of individual anthropogenic influences on these changes.

2. L21: *How about dried instead of “baked”?*

Corrected.

3. L73: *consistent instead of “consistently”?*

Corrected.

4. L87: *should “in the Northern Hemisphere” be omitted?*

Corrected.

5. L125: *Describing the regions in the figure 3 caption or in the main text (or at least refer to methods) would be helpful for the readers.*

As suggested, we have added a description of the monsoon region in the caption of Figures 3 and Supplementary Figs. 14-15.

6. L139: *illustrates instead of “illustrating”?*

Corrected.

7. L152: *How about the margin between the reduced and increased precipitation ..., instead of “the junction ...”?*

Corrected.

8. L165: *is it wet-to-dry or dry-to-wet here? Based on the previous sentence and Fig. S13, I believe it should be the latter.*

Due to the content and text revisions following other reviewers, this sentence has been removed in the revised manuscript.

9. L166: *The section “For the other four monsoon regions,” makes this sentence confusing. Consider omitting or rewording.*

Due to the content and text revisions following other reviewers, this sentence has been removed in the revised manuscript.

10. L171: *has been increasing, instead of “have been increasing”?*

Corrected.

11. L189: *Refer to Fig. S12 here.*

Corrected.

12. L198: Arrows in Fig. 6 and Fig S14 are barely visible. Increasing their size would be helpful.

Based on the suggestions, we have improved the quality of the figures, including making the arrows more visible.

13. L198-202: The sentence seems too long. How about removing “and” in L201 and starting a new sentence?

We have revised this sentence as suggested.

14. L233: Reference 44 is incomplete (L551).

Corrected.

15. L237: Please consider removing “, etc.,”.

We have removed “, etc.,” as suggested.

16. L240-245: As the first comments, I recommend rewriting this part to reflect the fact that the authors assessed the influence of individual forcings in CESM1 simulations, not in observations/reanalyses

Based on the reviewers' suggestions, we have rewritten this section to reflect what analyses we did with the different datasets separately and to avoid the misuse of D&A analysis. See Lines 314-320.

Using various precipitation datasets obtained from observations, reanalyses, a large multi-model ensemble (CMIP6), and a large single-model ensemble (CESM-LENs), we characterize the global occurrence frequency, transition duration and intensity of precipitation whiplash over the historical (1979-2019) period on the regional and global scales. We also analyze projected future changes in whiplash under the scenario of the RCP8.5 emissions using simulations and assess the influence of individual anthropogenic forcings using CESM-XLENs.

17. L249-253: I suggest moving this section to the end of the discussion section (at L271).

As suggested, we have moved this section to the end of the discussion section.

18. L340-341: Do you think the results would be different if you first remove the seasonal cycle and the trend from precipitation data, prior to calculating the whiplash events?

In our study, before calculating the whiplash events, we first detrended the raw precipitation data, then calculated the cumulative precipitation within 30 days to represent sub-seasonal precipitation, and finally removed the annual cycle of the cumulative precipitation for standardization. We tested the effect of several different operation sequences on our results. See the following Figure 1 for details.

We tested three approaches, *i*) detrending the raw data, then calculating the cumulative precipitation (grey filled areas), and finally removing the annual cycle (red lines) of the cumulative precipitation to calculate the standardized precipitation anomalies (black lines), See Figure 1a. *ii*) calculating the cumulative precipitation first, then detrending and removing the annual cycle of cumulative precipitation, See Figure 1b. *iii*) detrending the raw data and calculating the precipitation climatology for each calendar day (annual cycle), and then calculating the cumulative precipitation and multi-year mean cumulative precipitation, See Figure 1c.

The results of these three approaches are consistent, except that the third method results in missing precipitation for the first 29 days in the calculation of the cumulative value of the annual cycle. Thus, changing the sequence of the steps for preprocessing precipitation data has no effect on our results, and we choose the first approach. We have corrected and detailed our description in the revised manuscript. See Methods.

Figure 1 An example comparing the effects of different precipitation data pre-processing steps on the results.

19. L348: Should it be “lower and higher” instead of “lower (higher)”?

Corrected.

20. L351: Should wet-to-dry and dry-to-wet be swapped?

Corrected.

21. L383: is the part “(10th or 5th)” necessary?

Due to the content and text revisions following other reviewers, this sentence has been removed in the revised manuscript.

Reviewer #2

General comments:

In my opinion, this manuscript is well written and organized with good quality figures to present several important findings: i) precipitation whiplash in the current climate and projected future climate, ii) concurrent changes in precipitation total and whiplash, iii) attribution of precipitation whiplash changes to different anthropogenic forcings, and iv) large-scale atmospheric condition for precipitation whiplash in the current and projected future climate. Despite the vast results with multiple model simulation sets, the results are concisely and effectively presented. I think this manuscript deserves to be published in Nature Communications after minor revisions. The specific suggestions for improving the manuscript are as follows.

We wish to sincerely thank Reviewer #2 for his/her extensive and thoughtful comments which helps to improve our manuscript. We have revised the original manuscript (OM) following suggestions of Reviewer #2. There have been textual changes in the manuscript, mostly in sections of Results, Discussion and Conclusions, and Methods. The response to each point raised is given alongside the relevant specific comment below. Throughout, *reviewer comments* are in *blue* font and *italic* type, and our point-by-point response is in black font.

Minor revisions recommended:

1. This manuscript uses reanalysis precipitation products (ERA5, JRA55, and MERRA2) as observational precipitation data. I think at least one satellite-based precipitation product should be added to the analysis because reanalysis precipitation products include errors from model simulations.

As suggested, in the revised manuscript, we have added one satellite-based and two ground-based precipitation products to our analysis. The information table for the datasets is shown below, and we have added detailed description of datasets in Methods and Supplementary Table 2. In the revised manuscript, we compared observed and simulated climatology (See Fig.1, Supplementary Figs. 6-7 and corresponding analysis) and trends (See Fig.2, Supplementary Figs. 9-11 and corresponding analysis) of precipitation whiplash.

Supplementary Table 2. Summary of the six gridded precipitation datasets used in this study. Abbreviations in the data source(s) column are defined as follows: G, gauge; S, satellite; and R, reanalysis.

	Dataset	data source(s)	Spatial resolution	Coverage	Period used
1	ERA-5	R	0.25° × 0.25°	Global	1979-2019
2	MERRA-2	R	0.5° × 0.625°	Global	1980-2019
3	JRA-55	R	1.25° × 1.25°	Global	1979-2019
4	CHIRPS	G, S, R	0.05° × 0.05°	50° S-50° N Land only	1981-2019
5	GPCC	G	1° × 1°	Land only	1982-2019
6	REGEN_LongTermStns	G	1° × 1°	Land only	1979-2016

2. [L24] *It would be good to use the word “nitrogen” instead of “N” to avoid ambiguity.*

Corrected.

3. [L47] *Remove a typo “.”*

Corrected.

4. [L87-89] *It seems that this sentence should be revised because there is redundant information for regions.*

Corrected.

5. [L111] *“Fig. 2c-d” might be “Fig. 2a-b”.*

Corrected.

6. [L139] *“illustrating” might be “illustrates”.*

Corrected.

7. [L148] *Which land region do you mention? Do you mean the Arctic land region? Please clarify it.*

Due to the content and text revisions following other reviewers, this sentence has been removed in the revised manuscript.

8. [L165] “wet-to-dry” might be “dry-to-wet”.

Due to the content and text revisions following other reviewers, this sentence has been removed in the revised manuscript.

9. [L200] “western subtropic Pacific high (WPSH)” might be “western Pacific subtropical high (WPSH)”.

Corrected.

10. [L224-225] *It would be great if the authors could explain a little bit more about why the circulation anomalies will be deeper and more extensive in the future scenario. I think it would be an important step forward for better understanding the underlying mechanisms in changes in precipitation whiplash.*

As suggested, we have added a discussion about possible causes of future circulation anomalies changes. See lines 297-312.

In the future scenario (2060-2099; Figs. 6c-f and Supplementary Fig. 21c-f), the transitions of large-scale circulation patterns associated with dry and wet extremes are similar to the current period, suggesting that the atmospheric circulation patterns dynamically drive the occurrence of precipitation whiplash in the NEC. The positive and negative circulation anomalies are deeper and more extensive, implying a potentially stronger intensity and wider impact of precipitation whiplash. These more drastic and rapid swings between the dry and wet extremes could also be related to the enhanced seasonal hydrological cycle⁴⁹ due to an increased water holding capacity of the atmosphere in a warming climate⁵⁰. In addition, exploring the change in influencing factors moderating the circulation anomalies in future scenario, including sub-seasonal variability of WPSH and its possible influencing mechanisms (e.g., SSTs⁵¹), the amplitude of mid-latitude Rossby wave trains (East Asia-Pacific teleconnection pattern in meridional direction⁴¹ and Silk-Road pattern in latitudinal direction⁵²), large-scale teleconnection activities^{53,54}, and local positive feedback processes such as cloud properties and surface energy⁵⁵ may be capable of explaining the future variability of precipitation extremes and whiplash in the NEC region, which requires further work.

11. [L251] *Could you explain a little bit more what novel regional precipitation regimes are?*

As suggested, we have refined the description of “novel regional precipitation regimes” in the revised manuscript. See lines 388-393.

While humans are adapting to regional regimes of scarce or excess precipitation, the ongoing changes in precipitation regimes, including more frequent, intense and rapid dry-wet transitions resulting from the increasing temporal variability of sub-seasonal-scale precipitation, will complicate the water resources management and disaster prevention for human society. Natural ecosystems could also be considerably impacted by this novel regional precipitation regime.

12. [Discussion and Conclusions section] *The authors discussed atmospheric circulation conditions for extreme events in the current climate. However, one of the main objectives of this study is to investigate future changes in extreme events. So, I think the author should also discuss the future changes in atmospheric circulation conditions for extreme events.*

As suggested, we added a discussion on the future changes in precipitation regimes, extremes, variability and their driving causes including circulation factors. See lines 363-387.

In a future warming atmosphere, the tropospheric water vapor increases approximately following the Clausius-Clapeyron relationship^{50,59,60}, which influences the spatial pattern of changes in precipitation regimes in thermodynamic in addition to dynamic ways^{50,61,62}. On the one hand, increased atmospheric water vapor resulting from atmospheric warming promotes negative (positive) anomalies in moisture convergence in climatologically dry (wet) regions where downward (upward) motion is predominant, creating unfavorable (favorable) conditions for processes that trigger precipitation. On the other hand, change in precipitation alters the release of latent heat and thus the corresponding vertical motion, and further alters precipitation through this dynamic effect associated with changes in circulation^{50,61}. A pattern of increasing wetting in wet areas and drying in dry areas is found in both observations and simulations⁶³⁻⁶⁷. The warming effect not only leads to changes in the mean^{59,68} and extreme^{69,70} precipitation characterized by strong spatial heterogeneity, but also has a significant influence on precipitation variability^{7,71}. For example, the seasonal variability in global monsoon precipitation increases with global

warming and is mainly related to increased moisture convergence and surface evaporation⁷². In subtropical North America, the increase in sub-seasonal precipitation variability due to increased background moisture is offset by the effect of weakening circulation variability¹⁵, which is consistent with our projection that there is no significant change in whiplash events in the North American monsoon region. Therefore, projected changes in the characteristics of precipitation whiplash events are inseparable from changes in atmospheric moisture and circulation in a warming future climate context. Quantifying how thermodynamic and dynamical changes in a warmer climate exacerbate or weaken the whiplash between regional sub-seasonal-scale dry and wet extremes from the moisture budget perspective is practical for deeper analysis to understand the underlying mechanisms of precipitation whiplash.

13. [L305] *Could you describe a little bit more what natural forcing factors are and what other external anthropogenic factors are?*

As suggested, we added description of natural forcing factors and other external anthropogenic factors in the revised manuscript. See lines 438-444.

The CESM-XLENS simulations were conducted using the same configuration and initialization protocols as the original CESM-LENS experiment³³, except keeping the AER, GHG, and BMB conditions fixed at the level of 1920, respectively³⁴, while natural forcing factors (i.e., solar and volcanic), and all other external anthropogenic forcings (i.e., stratospheric and tropospheric ozone, land use/land cover changes and individual forcings other than the one that was fixed in 1920) follow historical and RCP 8.5 scenarios as the original CESM-LENS experiment did³³.

14. [L323-324] *In the context, precipitation extremes are firstly identified in native resolution, and then the results were re-gridded to a uniform 2-degree resolution. However, many previous studies perform re-gridding at the very first and then proceed following analyses. Do you have any specific reasons to do re-gridding after identifying precipitation extremes? Have you compared the results from those different processes?*

In our analyses, we re-grid the raw precipitation data first, and then proceed following analyses. In the original manuscript, "precipitation outputs" refers to the original outputs of precipitation dataset. We have rewritten it.

The above raw precipitation data are also re-gridded to a uniform 2° grid.

15. [L348] Please explain a little bit more how you processed detrend and annual-cycle-removing at the subseasonal scale and how you defined the subseasonal scale.

Before identifying precipitation extremes and whiplash, we preprocess the raw precipitation data.

1). We detrend the original precipitation data to remove the effects of long-term climate change. Taking CESM-LENS as an example, we use a linear fit to the annual mean precipitation of the historical radiative (1920-2005) and the RCP8.5 scenarios (2006-2100) data for each grid point, and then separate the short-term precipitation variability from the long-term climate change by subtracting its corresponding fitted trend value from the daily precipitation data for each year. We compare three different detrending methods: simple linear detrending of two series, polynomial (binomial) fit detrending of the entire time series (historical + RCP8.5), and a "scaling" method that estimates the trend based on regression of model ensemble mean and individual ensemble members (It is assumed here that the forced response can be estimated by averaging over model members ¹). The precipitation data (Fig. 1c in this response text) and whiplashes (Fig. 1d-e) using different detrending methods present very similar results. To be able to apply to the "observation" dataset, we choose the linear detrending method.

2). Cumulative precipitation totals (precipitation at sub-seasonal scale) are then calculated from a 30-day rolling sum of detrended daily precipitation over the entire period (black line in Fig. 2a in this response text). Annual-cycle-removed cumulative precipitation totals (black line in Fig. 2b) are calculated by Equation 1.

$$P_{ij}' = \frac{P_{ij} - \overline{P_j}}{\sigma_j} \quad (1)$$

where P_{ij} is the cumulative precipitation totals of the j^{th} ($j = 1, \dots, 365$) Julian day in year i (i

= 1920, ..., 2100), \bar{P}_j is the 181-year (1920-2100) mean of the j^{th} Julian day, and σ_j is the 181-year standard deviation of the j^{th} Julian day.

After above processing, the detrended and annual-cycle-removed precipitation totals at the sub-seasonal scale for each grid lower (higher) than the lower (upper) threshold values are defined as dry (wet) extreme events (Fig. 2b-c).

Overall, we define "precipitation at sub-seasonal scale" as 30-day cumulative precipitation. We detrend the raw precipitation data and then remove the annual cycle for sub-seasonal-scale precipitation. To avoid ambiguity, we reorganized the sentences in the revised manuscript. See Methods.

Figure 1. Global (a) annual mean precipitation (mm) and (b) precipitation trend (%) over 1920-2100 in the CESM-LENS ensemble. (c), Change in original (blue), polynomial detrended (green), ensemble mean scaled (yellow) and linear detrended (orange) annual mean precipitation, shaded areas indicate the encompasses 90% of the full 40-member CESM-LENS ensemble spread. (d-e), the occurrence frequency of (d) dry-to-wet and (e) wet-to-dry whiplash calculated by original (blue), polynomial detrended (green), ensemble mean scaled (yellow) and linear detrended (orange) precipitation.

Figure 2 An example illustrating the principle of the whiplash indices. **a**, the black (green) line indicates the detrended (raw) 30-day cumulative precipitation totals, the blue line indicates the annual cycle of cumulative precipitation totals, and the gray shading indicates the raw daily precipitation data. **b**, the standardized detrended (black) and raw (green) cumulative precipitation anomalies, and the yellow (green) dashed line indicates the 10th (90th) threshold over the current period (1979-2019; Methods). **c**, an example of dry-wet whiplash and the characteristics we are interested in. The light gray fill indicates the detrended 30-day cumulative precipitation

and the dark gray fill indicates the raw daily precipitation data.

16. [L349] “lower and upper” might be “lower (upper)”

Corrected.

17. [L361-367] This might not fit in the method section.

As suggested, we have moved these sentences to the section of Results.

18. [L360-361] I think more explanation is needed here. How did you average dates (1/1, 1/2, ... 12/31)?

We define the average timing of the events by the average date on which events have occurred. We calculate the average day within a year on which whiplash have occurred during the period of interest² for each grid. We first convert the date of occurrence D_i of a whiplash event in year i into an angular value θ_i by

$$\theta_i = D_i \cdot \frac{2\pi}{m_i} \quad 0 \leq \theta_i \leq 2\pi \quad (1)$$

where $D_i = 1$ corresponds to January 1 and $D_i = m_i$ to December 31, and where m_i is the number of days in that year. The average date of occurrence \bar{D} of whiplash at a grid is then defined as:

$$\bar{D} = \begin{cases} \tan^{-1}\left(\frac{\bar{y}}{\bar{x}}\right) \cdot \frac{\bar{m}}{2\pi} & \bar{x} > 0, \bar{y} \geq 0 \\ \left[\tan^{-1}\left(\frac{\bar{y}}{\bar{x}}\right) + \pi \right] \cdot \frac{\bar{m}}{2\pi} & \bar{x} \leq 0 \\ \left[\tan^{-1}\left(\frac{\bar{y}}{\bar{x}}\right) + 2\pi \right] \cdot \frac{\bar{m}}{2\pi} & \bar{x} > 0, \bar{y} < 0, \end{cases} \quad (2)$$

with

$$\bar{x} = \frac{1}{n} \sum_{i=1}^n \cos(\theta_i) \quad (3)$$

$$\bar{y} = \frac{1}{n} \sum_{i=1}^n \sin(\theta_i) \quad (4)$$

$$\bar{m} = \frac{1}{n} \sum_{i=1}^n m_i \quad (5)$$

where x (y) is the cosine (sine) components of the average date, and n is the total amount of whiplash at that grid.

As suggested, we have placed the description of the method of averaging the occurrence dates in Methods (**Climatology of the occurrence frequency and timing of precipitation whiplash**).

19. [L373, L380] “Fig. 2c-f” might be “Fig. 2c-d”.

Corrected.

20. [L398] In my understanding, F_i is an experiment without forcing i . If so, the text should be revised to clarify it.

Corrected.

21. [L1 in Fig.1 caption] “precipitation whiplash” might be “precipitation extremes and whiplash”.

Corrected.

22. [Fig. 2 caption] The information in parentheses in L3-4 and L5-7 are redundant.

We have removed the redundant sentences

23. [L8 in Fig.3 caption] “CESM-LENS ensemble” might be “CESM-LENS ensemble and CMIP6 ensemble”

Corrected.

24. [Fig. 4] I think reorganizing the position of color bars will help better perceive the figure. It would be great if the authors could reorganize the color bars with 2x2 rows and columns as below.

morePmoreE(top left), morePfewerE(top right)

lessPmoreE(bottom left), lessPfewerE(bottom right)

As suggested, we have reorganized the position of color bars.

25. [Fig. 6] It is difficult to identify the directions of the arrows on the figure. I think the quality of the arrows should be improved.

As suggested, we have improved the quality of the figures, including making the arrows more visible.

#Reviewer 3

Review of the manuscript entitled “Increasing global precipitation whiplash due to anthropogenic greenhouse gas emissions” for Nature Communications

The paper aims at detecting and attributing changes in subseasonal precipitation “whiplash” over the 1920-2100 period, based on a large ensemble of CESM simulations with all or individually-removed anthropogenic forcings and a CMIP6 multi-model ensemble (54 realizations of 22 models). Anthropogenic GHG emissions are found to dominate the projected changes, with a 20% global increase in the whiplash occurrences that outpace the effect of the anthropogenic aerosols (AER) after the 2020s. A regional analysis is also conducted over six monsoon regions. Finally, the authors claim that the subseasonal shifts in precipitation are associated with shifts in circulation patterns, as illustrated by a composite analysis over Northeast China (NEC).

The paper proposes an original study based on a large number of data sets, but it suffers from several weaknesses and does not add much value to previous studies that have already quantified an increase in rainfall variability at multiple timescales (e.g., Pendergrass et al., 2017). I therefore recommend the study to be published in another journal or to be resubmitted to Nature Communications following some of the following suggestions. Please, find hereafter my major comments as well as a brief (non-comprehensive) list of more specific comments.

We wish to sincerely thank Reviewer #3 for his/her extensive and thoughtful comments which helps to improve our manuscript. We absolutely agree that changes in precipitation variability at various temporal scales have been intensively investigated. However, what changes in precipitation variability imply to the detailed temporal transitions of precipitation extreme regimes (wet and dry) remains unexplored. We believe our results can provide more insights into changes in precipitation regimes under a changing climate, which could be linked to changes in precipitation variability.

We have revised the original manuscript (OM) following the suggestions of Reviewer #3. There have been textual changes in the manuscript, mostly in sections of Results, Discussion and

Conclusions, and Methods. The response to each point raised is given alongside the relevant specific comment below. Throughout, *reviewer comments* are in *blue* font and *italic* type, and our point-by-point response is in black font.

Major comments:

1) Detection-attribution:

While the authors claim in their abstract that they “detect and attribute changes in precipitation whiplash”, they just assess the effect of individual anthropogenic forcings in the CESM model. As defined more than a decade ago by an IPCC guidance paper (Hegerl et al., 2010), “detection of change” is defined as the process of demonstrating that climate has changed in some defined statistical sense and, therefore, primarily relies on observations. Here, no observational dataset is used. Three global atmospheric reanalyses are used to evaluate the 1980-2014 climatology of the CESM and CMIP6 models in Fig. 1, but these are not observations and there is no attempt to perform a formal D&A analysis (reanalyses are not even shown in subsequent Fig. 2, 3, 5 and 6). If the authors decide to resubmit this study, I would thus suggest to focus the analysis over land (given the motivations), to use gridded observations (e.g., REGEN) in addition to reanalyses, and to perform a formal D&A study (or at least to compare simulated and observed trends).

1). Based on the reviewers' suggestions, we have reworded “*detect and attribute*” in the revised manuscript to avoid the misuse of D&A analysis. We clarified in the revised manuscript that we one of our works is assessing the effect of individual anthropogenic forcings in the CESM model. e.g.,

Lines 5-7: We detect observed and projected changes in characteristics of sub-seasonal precipitation whiplash and investigate the role of individual anthropogenic influences on these changes.

Lines 314-320: Using various precipitation datasets obtained from observations, reanalyses, a large multi-model ensemble (CMIP6), and a large single-model ensemble (CESM-LENS), we characterize the global occurrence frequency, transition duration and intensity of precipitation

whiplash over the historical (1979-2019) period on the regional and global scales. We also analyze projected future changes in whiplash under the scenario of the RCP8.5 emissions using simulations and assess the influence of individual anthropogenic forcings using CESM-XLENS.

2). As suggested, in the revised manuscript, we have added one satellite-based and two ground-based precipitation products to our analysis. The information table for the datasets is shown below, and we have added detailed description of datasets in Methods and Supplementary Table 2. In the revised manuscript, we compared observed and simulated climatology (See Fig.1, Supplementary Figs. 6-7 and corresponding analysis) and trends (See Fig.2, Supplementary Figs. 9-11 and corresponding analysis) of precipitation whiplash.

Supplementary Table 2. Summary of the six gridded precipitation datasets used in this study. Abbreviations in the data source(s) column are defined as follows: G, gauge; S, satellite; and R, reanalysis.

	Dataset	data source(s)	Spatial resolution	Coverage	Period used
1	ERA-5	R	0.25° × 0.25°	Global	1979-2019
2	MERRA-2	R	0.5° × 0.625°	Global	1980-2019
3	JRA-55	R	1.25° × 1.25°	Global	1979-2019
4	CHIRPS	G, S, R	0.05° × 0.05°	50° S-50° N Land only	1981-2019
5	GPCC	G	1° × 1°	Land only	1982-2019
6	REGEN_LongTermStns	G	1° × 1°	Land only	1979-2016

2) Whiplash definition:

While the concept of “precipitation whiplash” is original and may deserve a particular attention, the motivations are not entirely convincing and the definition raises a number of questions (so that those events may correspond to different phenomena across different regions or

models). A whiplash is defined as an abrupt shift between contrasted precipitation extremes, which means that the authors first need to define precipitation extremes. Their definition is based on the exceedance (deficit) of precipitation accumulated over 30 consecutive days above (below) the 90th (10th) preindustrial percentile. A linear detrending and annual cycle removing is applied to the 30-day running mean precipitation timeseries before identifying extremes in order to eliminate the effect of precipitation trends and seasonality. The motivations highlight potential impacts of such abrupt shifts on ecosystems, erosion or water quality, but seems to ignore that the succession of two extremely wet or two extremely dry months may have even stronger impacts. The authors claim (but actually don't show) that their results are not sensitive to the number of consecutive days (within the 20 to 40-day interval), but some of the impacts that motivate their study may occur at either shorter or longer timescales, given the asymmetry of wet versus dry extremes (short-duration heavy precipitation events can be enough to trigger large impacts). Beyond the selected 30-day timescale (which does not add much value to using daily instead of monthly data and raises the question of selecting independent extreme events given the use of running mean daily precipitation), there are other arbitrary choices that may need further justifications or explanations. Linear detrending is not necessarily the most suitable choice to remove changes in mean precipitation given the time-varying effect of AER and the availability of multiple realizations (at least this could be checked with CESM). Changes in precipitation variability have been already documented (see for instance the ARG6 WG1 assessment from both Chapter 8 and 11) so that the focus could be more specifically on "whiplash occurrences" by computing time-varying (rather than fixed preindustrial) percentiles. So doing, the number of subseasonal wet/dry extremes would not change over time, but the shifts could still change (in terms of both occurrences and seasonality, cf. Fig.1) thereby highlighting an original result beyond the increase in variability. Finally, the distinction of "wet to dry" versus "dry to wet" transitions is so far not fully justified given the apparent similarity between their climatologies (Fig. 1) and responses (Fig. 2), except over the WafriM region (cf. Fig. 3) which may need further explanations. This strong symmetry between the two types of precipitation whiplash may be linked to their definition and may suggest alternative

definitions where wet extremes do not necessarily last as long as the selected dry extremes (here 30 days). In this case, and beyond changes in occurrences and seasonality, other changes in these events could be analysed such as the mean duration of wet extremes and the mean precipitation budget across such transitions (increasing excess or deficit?). If the authors decide to resubmit this study, I would thus suggest to use another and potentially more fruitful definition of a subseasonal precipitation whiplash.

Previous literature related to the rapid transition between dry and wet states can be roughly divided into three categories. *i*) Most of the previous studies assessing future changes in precipitation variability at the global scale presented results only for simple indices³⁻⁸, such as changes in the standard deviation of precipitation^{6,7}, interannual variability of precipitation P minus evaporation E^3 , or differences between dry and wet seasons⁴, and assess an overall future trend at different scales^{6,7}. *ii*) Global studies at seasonal scales mostly use monthly data longer than 3 months, such as 3-month running average precipitation⁴, or 3-month SPEI⁵. Fewer studies have addressed precipitation variability at sub-seasonal scales using daily precipitation datasets. *iii*) Most of the studies analyzing the mechanisms of rapid precipitation shifts are based on a particular country^{9,10} or region¹¹⁻¹⁴ and focus on a specific event of interest¹⁰⁻¹⁴ (most of them refer to the transition between a specific extreme dry year and an extreme wet year).

However, what changes in precipitation variability imply to the detailed temporal transitions of precipitation extreme regimes (wet and dry) remains unexplored. Based on the above considerations, we have improved the whiplash indices we defined in the original manuscript as suggested by reviewer #3. We believe that our defined precipitation whiplash indices can complement the details of precipitation variability changes by specific characteristics (e.g., frequency, intensity, and transition duration of whiplash events) to provide more insights into the changes in precipitation regimes under a changing climate.

Note that the results based on our revised indices do not change the original conclusions (e.g., overall trends and human-driven effects) compared to the original manuscript. The events calculated by the revised and detail-processed whiplash indices differ only in the magnitude from

the results and conclusion presented in the original manuscript. For the original whiplash indices, we only defined the occurrence of transitions. In the modified version, we have analyzed additional innovative details, such as the transition duration of precipitation whiplash, and the intensity of transitions.

1) We spliced some dry or wet extremes that are briefly interrupted and then continued within a reasonable range based on a method similar to run theory¹⁵, and ensured the length of the events (events that appeared only briefly were removed; see the Methods section for a detailed description). As a result, we selected events that maintained a dry or wet state for a period of time and were of research interest, essentially overcoming the problem of selecting independent events as suggested by the reviewer#3. Furthermore, we recorded the length of dry and wet extremes as filled with yellow (dry) and green (wet) colors in Figure 1 below in this response text, so that events similar to the two consecutive months of extreme wet or dry (if any) proposed by the reviewer can also be retained in our definition (similar to extremes of longer duration in our definition).

Figure 1 An example illustrating the principle of the whiplash indices. **a**, the black (green) line indicates the detrended (raw) 30-day cumulative precipitation totals, the blue line indicates the annual cycle of cumulative precipitation totals, and the gray shading indicates the raw daily precipitation data. **b**, the standardized detrended (black) and raw (green) cumulative precipitation anomalies, and the yellow (green) dashed line indicates the 10th (90th) threshold over the current period (1979-2019; Methods). **c**, an example of dry-wet whiplash and the characteristics we are interested in. The light gray fill indicates the detrended 30-day cumulative precipitation and the dark gray fill indicates the raw daily precipitation data.

2) Compared to previous studies using monthly data (e.g., Pendergrass, et al. ⁷), we believe that the approach we defined is able to capture the occurring events more in detail. For example, if the dry-to-wet event occurs in April, the monthly data may only record March as dry and April as wet, which may lose some details and may also allow the sharp transition in precipitation to be averaged out during the month in which the transition occurs. However, our method allows for a more detailed recording of the exact time, duration, intensity, and transition duration (and thus the speed of transition) of the event (Figure 1 in this text).

3) As suggested, we added figures about sensitivity analysis on the number of consecutive days, extreme thresholds, etc. to the revised manuscript and demonstrated that our results are robust (See Methods and Supplementary Figs. 4-5).

4) Since we are concerned with wet and dry extreme events and their transitions on a sub-seasonal scale, we need to use a fixed number of consecutive days. Without considering consecutive precipitation, the events of interest are short-lived intense precipitation and droughts on the weather scale (in fact, such events generally break after a few days). The resulting short-term but intense extremes are not in our consideration from a sub-seasonal scale perspective, so we set a fixed number of consecutive days.

We have added a discussion on the limitations of our focus on cumulative precipitation at sub-seasonal scales. See lines 396-403.

A potential limitation is that our method of using sliding windows to accumulate sub-seasonal precipitation may smooth out short-lived, intense rainstorm events that can cause flooding, transportation disruptions, damage to urban infrastructure and loss of life⁷³. Events such as short-duration intense rainstorms following a prolonged drought that pose a significant threat to the emergency response system can be explored at more flexible spatiotemporal scales in subsequent studies. Recognizing the all-around risk induced by more volatile precipitation and more whiplash under multiple timescales is the first step to develop a resilient coupled natural and human system in a warming climate.

5) Based on the reviewer's suggestion, we considered a time-varying threshold approach (where all conditions except the threshold do not change) to calculate the precipitation whiplash, but a problem is that using a time-varying threshold makes it difficult to compare the results in future scenarios of all radiative forcing and different single forcing under the same threshold.

6) We detrend the original precipitation data to remove the effects of long-term climate change. Taking CESM-LENS as an example, we use a linear fit to the annual mean precipitation of the historical radiative (1920-2005) and the RCP8.5 scenarios (2006-2100) data for each grid point, and then separate the short-term precipitation variability from the long-term climate change by subtracting its corresponding fitted trend value from the daily precipitation data for each year. As suggested, we compare three different detrending methods: simple linear detrending of two series, polynomial (binomial) fit detrending of the entire time series (historical + RCP8.5), and a "scaling" method that estimates the trend based on regression of model ensemble mean and individual ensemble members. The "scaling" method assumes that the forced response can be estimated by averaging over model members ¹. The precipitation data (Fig. 2c) and whiplashes (Fig. 2d-e) using different detrending methods present very similar results. To be able to apply to the "observation" dataset, we choose the linear detrending method.

Figure 2. Global (a) annual mean precipitation (mm) and (b) precipitation trend (%) over 1920-2100 in the CESM-LENS ensemble. (c), Change in original (blue), polynomial detrended (green), ensemble mean scaled (yellow) and linear detrended (orange) annual mean precipitation, shaded areas indicate the encompasses 90% of the full 40-member CESM-LENS ensemble spread. (d-e), the occurrence frequency of (d) dry-to-wet and (e) wet-to-dry whiplash calculated by original (blue), polynomial detrended (green), ensemble mean scaled (yellow) and linear detrended (orange) precipitation.

3) CESM versus CMIP6 ensembles

The authors make use of the agreement between two ensembles to highlight the reliability of their results (e.g., L94-96). So doing they seem to ignore the different nature of the two ensembles

despite a reasonable description in the Methods section.

The CESM-LENS large ensemble is aimed at quantifying the contribution of internal climate variability to projected changes in key variables, while the CMIP6 ensemble (especially the way it is used in the present study) allows the authors to document the inter-model spread. The fact that CESM is generally relatively close to the CMIP6 multi-model ensemble (MME) mean does not necessarily mean that the results are reliable, especially if the CMIP6 MME spread is substantial (it only means that CESM is fairly representative of the CMIP6 MME mean climatology or response).

The stippling/hatching (hatching only could be enough) shown in Fig. 2 and 4 only shows the statistical significance of the changes projected by CESM (compared to internal variability) and the global maps should be shown for CESM and CMIP6 separately. Moreover, the captions are also not clear about the shading around the CMIP6 MME mean response in both Fig. 2 and 3. Should we conclude that internal climate variability within CESM is as strong as the inter-model spread within CMIP6?

If the authors decide to resubmit this study, they may want to pay more attention to the signal-to-noise ratio and use CESM to diagnose what is the minimum ensemble size needed to detect simulated changes in whiplash occurrences (in a high-emission scenario, a larger size would be probably needed for lower-emission scenarios). Then they could select a subset of CMIP6 models that satisfy this criterion in order to assess the inter-model spread and discuss the reliability of their results.

Referring to reviewer #3's suggestion, we are aware of the need to consider uncertainties in climate projections when using global climate models (GCMs) to quantitatively project future changes in weather or climate events of interest^{16,17}. For example, internal climate variabilities may mask the signal of human-induced climate change, and separating anthropogenic climate change from the internal climate variability is essential to providing reliable projections and better estimates of the contribution of anthropogenic impacts to projected climate change. The practical capacity of climate projection depends largely on the signal-to-noise ratio of the projections, that

is, how much the expected climate change is compared to the internal climate uncertainty in the projections^{18,19}.

In the original manuscript, we used two large ensembles (CESM-LENS and CMIP6) to quantify the predicted variability of precipitation whiplash, although they have different nature. In a given radiative forcing scenario, the ensemble model uncertainty of CESM-LENS consists of internal climate variability only, while that of CMIP6 composed of a combination of internal climate variability and model formulation differences (i.e., structural uncertainty) with unclear relative importance^{1,20}.

Therefore, as suggested, in the revised manuscript, we use the CESM-LENS results as the estimate of the long-term trend of the precipitation whiplash and focus on the uncertainty of the climate projections and identify the timing of the emergence of anthropogenic signal of changes in precipitation whiplash through the signal-to-noise ratio. The CMIP6 ensemble is used for comparison with CESM-LENS to enhance the robustness of the results.

A prerequisite is that we need to determine that the spread of the CMIP6 ensemble is well representative of the internal variability generated by CESM-LENS. First, we use an approach similar to Kay, et al.²¹ to determine how much of the spread in the CMIP6 projections can be explained by internal climate variability alone. We assume that the CESM-LENS spread is across the range of internal variability. We calculate the trend in the frequency of precipitation whiplash for each member of CESM-LENS and CMIP6 in the current period (1979-2019) and future period (2060-2099), respectively, and then quantify the spread of frequency trends with the standard deviation of the trends across members. The *f* test is then used to evaluate whether the model spread of CMIP6 is statistically different from that of CESM-LENS. Figure 3 indicates that for the frequency of precipitation whiplash in most regions, regardless of the period, the trend spread estimated with CESM-LENS, which is generated by internal climate variability only, is not statistically significantly different from the trend spread within CMIP6 (areas not dotted in the CMIP6 panel in Figure 3), implying that the CMIP6 spread over most regions can be explained by the internal climate variability estimated by CESM-LENS. The spatial pattern of the standard

deviation of the annual mean trends within the two periods for the dry-to-wet and wet-to-dry whiplash events is similar for both ensembles. The magnitude of spread in the CMIP6 ensemble is larger (colors in the CMIP6 maps are darker) compared to CESM-LENS, which is consistent with the expectation that CMIP6 encompasses model variability and internal climate variability.

Figure 3 Global maps of standard deviation in frequency trends of whiplash for the (top two rows) current period (1979-2019) and (bottom two rows) future period (2060-2099). Trends are shown for both the 40-member CESM-LENS ensemble and the 55-member CMIP6 ensemble. Stippling on the CMIP6 maps indicates standard deviations that are statistically different than the CESM-LENS for the corresponding period. Stippling is based on an f test and a 95% confidence interval.

In addition, we calculate the ratio of the standard deviation of the trend between CESM-LENS and CMIP6 members derived above (CESM-LENS divided by CMIP6), to simply quantify the

contribution of internal variability to the spread of the trend in the CMIP6 ensemble, by referring to Deser, et al. ¹. A ratio greater (smaller) than 0.5 means that the contribution of internal variability is greater (smaller) compared to model variability, and a ratio > 0.75 means that the spread in the CESM-LENS trend is not significantly different from that in the CMIP6 trend. The results (not shown) show that for both whiplash events, internal climate variability is more important than structural differences of models in the current (future) period in 98% (95%) of the regions. In 83% (82%) of the regions, there is no significant difference between the model spread of these two large ensembles. Therefore, the above analysis supports that the spread of the CMIP6 ensemble is representative of the internal variability generated by CESM-LENS in our study. The CMIP6 subset we selected is suitable for use with CESM-LENS to enhance the robustness of the results.

Therefore, as suggested, we have made substantial revisions in the revised manuscript.

1). We use the CESM-LENS results as the estimate of the long-term trend of the precipitation whiplash and focus on the uncertainty of the climate projections. The CMIP6 ensemble is used for comparison with CESM-LENS to enhance the robustness of the results (Figs. 2, S9-11 and corresponding analysis).

2). Both CMIP6 and CESM-LENS are used to predict the response of future precipitation whiplash to external forcing (See Methods). We focus on the signal-to-noise ratio to diagnose the minimum number of ensemble members needed to detect the forced signal in both CMIP6 and CESM-LENS (Supplementary Figs. 12-13 and corresponding analysis).

3). We also estimate the timing of the emergence of anthropogenic signal of changes in precipitation whiplash globally and in six monsoon regions (Fig. 3, S12-15 and corresponding analysis).

In addition, we have revised the details according to the suggestions.

1). We showed the global maps of CESM-LENS and CMIP6 as well as the global and land mean results in Figure 2 and Figure S8-10, respectively.

2). We only retained hatching to show the statistical significance of the predicted changes.

3). We improved the description of the figure caption about the shading around the CMIP6

MME mean response in the related figure.

4) Role of shifts in large-scale circulation:

In a final subsection, the authors analyse the large-scale circulation background during precipitation whiplash, taking NEC as an example. They suggest a key role of moisture transport, which is not a big surprise. My feeling is that this section is not mature enough to be useful in the submitted manuscript. The authors recognize that this is a preliminary investigation and that the mechanisms leading to precipitation whiplash may differ among regions. They also conclude that such an analysis should be done in the context of circulation changes affecting specific regions.

If the authors decide to resubmit this study, they may need to choose between two options: 1) assess the role of large-scale circulation much more thoroughly and in the context of a specific region as a separate paper; 2) develop a more generic analysis framework, potentially applicable at the global scale, aiming at isolating the contribution of changes in large-scale circulation to changes in the occurrence frequency of precipitation whiplash. My feeling is that option 2 would be very ambitious so that option 1 may be more tractable and would allow the authors to focus more thoroughly on my previous comments before resubmitting their study.

Taking into account the comments of other reviewers, we decided to retain this subsection temporarily. We have added the importance of this subsection and further research goals (such as the moisture budget mentioned by reviewer #3) in the Discussion and Conclusions section.

See Lines 341-345: *To further understand the mechanisms underlying changes in precipitation whiplash, we link the patterns and transition processes of circulation anomalies to dry and wet extremes during whiplash events. We explore the variability of large-scale processes during precipitation whiplash in the six monsoon regions mentioned, despite only showing detailed results for the NEC region here as an example.*

See Lines 381-387: *Therefore, projected changes in the characteristics of precipitation whiplash events are inseparable from changes in atmospheric moisture and circulation in a warming future climate context. Quantifying how thermodynamic and dynamical changes in a warmer climate exacerbate or weaken the whiplash between regional sub-seasonal-scale dry and*

wet extremes from the moisture budget perspective is practical for deeper analysis to understand the underlying mechanisms of precipitation whiplash.

Specific comments: I also have a number of specific comments that I won't detail at this stage given the recommended rejection of the current manuscript. They include:

- the need to improve Figure captions;

We have rechecked and improved Figure captions.

- multiple needed edits to the text;

We have rechecked the manuscript.

- missing relevant references;

We have rechecked the references of the revised manuscript.

- lack of confidence intervals when providing global (or regional) estimates of the projected changes;

We have double checked the descriptions of confidence intervals in the graphs and figure captions, and have added confidence intervals to all descriptions related to the magnitude of change in the revised manuscript.

- the ensemble size of the CESM-XLENS simulations;

Because we only discussed the minimum number of CESM-LENS members needed to detect the forced signal, and did not select a smaller subset for our subsequent study. Similarly, we decided to retain all the ensemble members of CESM-XLENS.

- RCP8.5 vs SSP5-8.5 scenario;

As suggested, we have added a description of the emission levels for SSP5-8.5.

We used a multi-model ensemble consisting of 55 realizations of 22 distinct climate models (See Table S1) from CMIP6 project over the same period (i.e., 1920-2014 for the historical period and 2015-2100 for the SSP5-8.5 emission scenarios whose expected radiative forcing level in 2100 is 8.5 W m^{-2} similar to RCP8.5)

- a possible analysis of changes in whiplash seasonality (given the present-day climatology

shown in Fig. 1);

As suggested, we have added an analysis of changes in whiplash seasonality. See Supplementary Fig. 8 and Lines 103-110.

- a possible explanation of the lack of increase after 2060 (L102);

- a possible discussion of the limitations of the “dry-get-drier” paradigm;

- a possible explanation of the contrasted response shown in Fig. 3 over West Africa; - etc...

Due to the content and text revisions following other reviewers, these sentences have been removed in the revised manuscript.

References

- 1 Deser, C., Phillips, A., Bourdette, V. & Teng, H. Uncertainty in climate change projections: the role of internal variability. *Climate dynamics* **38**, 527–546 (2012).
- 2 Mardia, K. V. Statistics of directional data. *Journal of the Royal Statistical Society: Series B (Methodological)* **37**, 349–371 (1975).
- 3 Seager, R., Naik, N. & Vogel, L. Does global warming cause intensified interannual hydroclimate variability? *Journal of Climate* **25**, 3355–3372 (2012).
- 4 Chou, C. *et al.* Increase in the range between wet and dry season precipitation. *Nature Geoscience* **6**, 263–267 (2013).
- 5 Chen, H. & Wang, S. Accelerated transition between dry and wet periods in a warming climate. *Geophysical Research Letters* **49**, e2022GL099766 (2022).
- 6 Zhang, W. *et al.* Increasing precipitation variability on daily-to-multiyear time scales in a warmer world. *Science advances* **7**, eabf8021 (2021).
- 7 Pendergrass, A. G., Knutti, R., Lehner, F., Deser, C. & Sanderson, B. M. Precipitation variability increases in a warmer climate. *Scientific reports* **7**, 17966 (2017).
- 8 Cheng, L. & Liu, Z. Detectable Increase in Global Land Areas Susceptible to Precipitation Reversals under the RCP8.5 Scenario. *Earth's Future*, e2022EF002948 (2022).
- 9 Dong, L., Leung, L. R. & Song, F. Future changes of subseasonal precipitation variability in North America during winter under global warming. *Geophysical Research Letters* **45**, 12,467–412,476 (2018).
- 10 Kendon, M., Marsh, T. & Parry, S. The 2010–2012 drought in England and Wales. *Weather* **68**, 88–95 (2013).
- 11 Christian, J., Christian, K. & Basara, J. B. Drought and pluvial dipole events within the great plains of the United States. *Journal of Applied Meteorology and Climatology* **54**, 1886–1898 (2015).
- 12 Seager, R., Pederson, N., Kushnir, Y., Nakamura, J. & Jurburg, S. The 1960s drought and the subsequent shift to a wetter climate in the Catskill Mountains region of the New York City watershed. *Journal of Climate* **25**, 6721–6742 (2012).
- 13 Dong, X. *et al.* Investigation of the 2006 drought and 2007 flood extremes at the Southern Great Plains through an integrative analysis of observations. *Journal of Geophysical Research: Atmospheres* **116** (2011).
- 14 Swain, D. L., Langenbrunner, B., Neelin, J. D. & Hall, A. Increasing precipitation volatility in twenty-first-century California. *Nature Climate Change* **8**, 427–433 (2018).
- 15 Yevjevich, V. M. *Objective approach to definitions and investigations of continental hydrologic droughts*, An, Colorado State University. Libraries, (1967).
- 16 Meehl, G. A. *et al.* Decadal prediction: can it be skillful? *Bulletin of the American Meteorological Society* **90**, 1467–1486 (2009).
- 17 Masson-Delmotte, V. *et al.* Contribution of working group I to the sixth assessment report of the intergovernmental panel on climate change. *Climate Change 2021: The Physical Science Basis* (2021).
- 18 Hawkins, E. & Sutton, R. The potential to narrow uncertainty in regional climate predictions. *Bulletin of the American Meteorological Society* **90**, 1095–1108 (2009).
- 19 Hawkins, E. & Sutton, R. The potential to narrow uncertainty in projections of regional precipitation

- change. *Climate dynamics* **37**, 407-418 (2011).
- 20 Tebaldi, C., Arblaster, J. M. & Knutti, R. Mapping model agreement on future climate projections. *Geophys. Res. Lett.* **38**, L23701 (2011).
- 21 Kay, J. E. *et al.* The Community Earth System Model (CESM) large ensemble project: A community resource for studying climate change in the presence of internal climate variability. *Bulletin of the American Meteorological Society* **96**, 1333-1349 (2015).

REVIEWERS' COMMENTS

Reviewer #1 (Remarks to the Author):

I would like to thank the authors for the thorough revision and answers to reviewer comments. I am satisfied with the answers provided. I have some minor comments.

Overall comments:

The analyses are comprehensive, and the figures are of high quality. However, I had a hard time understanding some sentences. I have pointed out some below in line-by-line comments. Generally, I thought some sentences and paragraphs can be concise. Figure 4 caption does not clearly indicate what the figure shows.

Line-by-line comments:

L5: "detect" implies statistically distinguishing the signal from the internal variability in observations. I suggest removing this word.

L13: "...increased and decreased..." Add that this is regarding the historical simulations.

L14-16: projected to increase by when?

L130: remove the word "all"

L143: Robust instead of "robustness".

L224-227: This sentence seems to be stating the same message twice. Consider shortening.

L262: Remove the word "sandwich".

L302: Use "stronger" instead of "deeper and more extensive".

L366: "...thermodynamic in addition to dynamic ways": Not clear. Consider rewording the whole sentence.

L386: "from the moisture budget perspective..." : Not clear. Consider rewording the whole sentence.

L389: conditions instead of regimes?

L389-393: This sentence is too long and not clear. Consider breaking it up into multiple sentences.

L406-412: Is this paragraph necessary?

L430-431: Do you mean ensemble mean precipitation whiplash characteristics?

L431: How about "forced changes" instead of regimes?

L449: Remove the word "defined".

L467: By model mean do you mean the ensemble mean?

L467: Remove "weight equally".

L468-469: This sentence (the last sentence of the paragraph) is redundant.

L471: How about "To compare with" instead of "For comparison with"?

L472: Numbers of datasets (five, four, and four) in this sentence are not correct.

L476: Should be "The satellite-based dataset is".

L487: Instead of "annual-cycle-removed", try "annual cycle of the is removed."

L495: "conclusions are not".

L510-512: What do you mean here by "does not overlap temporally"?

L548: "corresponds to December 31"

L562: Relative change of what?

L575: While it is the internal climate variability for CESM-LENS, for CMIP6, the noise includes the intermodel variability as well. This should be noted here.

L586: Change to "... extremes, whiplash, and precipitation totals"

L585-591: This is mentioned in the figure 4 caption. Do you need this paragraph?

L592: Change to "Anthropogenic influence on precipitation whiplash"

L613: Removed "from the study days".

L623: SAsiaM is missing the second "s" throughout the manuscript.

L890: ")" is missing?

L897: The figure title says "transition duration". It should be changed to the occurrence frequency.

L911: This figure caption is not clear. Please consider rewriting it.

L916: Use darker instead of deeper.

L918: In the figure legends, Instead of "fewer E", how about "less E"?

L934: Replace "Average atmospheric anomalies".

Reviewer #2 (Remarks to the Author):

Recommendation: Accept after minor revisions

General comments:

The authors carefully addressed the concerns that were raised in my initial review and revised the manuscript for improvement. I think the revised manuscript is much improved and is worthy of publication in Nature Communications. However, I have found some typos as listed below, requiring careful checking of typos throughout the entire revised manuscript before submitting it.

Minor revisions recommended:

1. [L472] It does not seem that this study use "five reanalysis, four satellite-based datasets and four ground-based datasets". Please correct this.
2. [L298] "Figs. 6c-f" might be "Figs. 6d-f".
3. [L298] "Supplementary Figs. 21c-f" might be "Supplementary Figs. 21d-f".
4. [L530] "Annual-cycle-removed cumulative precipitation totals" might be "Standardized annual-cycle-removed cumulative precipitation anomalies".
5. [L555] "n is the total amount of whiplash at that grid" might be "n is the number of total years".

Reviewer #3 (Remarks to the Author):

My main requests have been addressed and the manuscript has been substantially improved compared to the original draft. I have therefore no more objection to the publication of the paper in Nature Communications.

Reviewer #1

I would like to thank the authors for the thorough revision and answers to reviewer comments. I am satisfied with the answers provided. I have some minor comments.

Overall comments:

The analyses are comprehensive, and the figures are of high quality. However, I had a hard time understanding some sentences. I have pointed out some below in line-by-line comments. Generally, I thought some sentences and paragraphs can be concise. Figure 4 caption does not clearly indicate what the figure shows.

We wish to sincerely thank you for your clear and detailed feedback which helps us to improve the manuscript. We have revised the manuscript following your suggestions. The response to each point raised is given alongside the relevant specific comment below. Throughout, *reviewer comments* are in *blue* font and *italic* type, and our point-by-point response is in black font.

Line-by-line comments:

L5: “detect” implies statistically distinguishing the signal from the internal variability in observations. I suggest removing this word.

As suggested, we have replaced "detect" with "quantify".

L13: “..increased and decreased....” Add that this is regarding the historical simulations.

As suggested, we have added " In historical simulations, " in this sentence.

L14-16: projected to increase by when?

As suggested, we have added "by 2079" in this sentence.

L130: remove the word “all”

Corrected.

L143: Robust instead of “robustness”.

Corrected.

L224-227: This sentence seems to be stating the same message twice. Consider shortening.

As suggested, we have shortened this sentence.

The results suggest that GHGs are projected to increase the occurrence frequency of precipitation whiplash and facilitate more violent transitions during whiplash events in many economically and demographically important regions of the globe by the end of the 21st Century.

L262: Remove the word “sandwich”.

Removed.

L302: Use “stronger” instead of “deeper and more extensive”.

Changed.

L366: “..thermodynamic in addition to dynamic ways”: Not clear. Consider rewording the whole sentence.

As suggested, we have reworded this sentence.

In a future warming atmosphere, the tropospheric water vapor increases approximately following the Clausius-Clapeyron relationship^{50,59,60}, which influences the spatial pattern of changes in precipitation regimes in both thermodynamic (moisture) and dynamic (circulation) ways.

L386: “from the moisture budget perspective...” : Not clear. Consider rewording the whole sentence.

As suggested, we have reworded this sentence.

Quantifying how thermodynamic and dynamical changes in a warmer climate exacerbate or weaken the whiplash between regional sub-seasonal-scale dry and wet extremes is practical for a comprehensive analysis to understand the underlying mechanisms of precipitation whiplash.

L389: conditions instead of regimes?

Precipitation regimes describe the long-term patterns and characteristics of precipitation in a given region or climate, and precipitation conditions usually refer to the short-term or immediate conditions that affect precipitation. In this context, our intended meaning in this sentence is that humans have been adapting to shifts in regional precipitation regime towards drier (scarce precipitation) or wetter conditions (excess precipitation). Therefore, I believe it is appropriate to retain the term "precipitation regime" in the manuscript.

L389-393: This sentence is too long and not clear. Consider breaking it up into multiple sentences.

As suggested, we have reworded this sentence.

Although humans are adapting to regional regimes of scarce or excess precipitation, this adaptation will be complicated by ongoing changes in precipitation regimes. The more frequent, intense and rapid dry-wet transitions resulting from the increasing temporal variability of sub-seasonal-scale precipitation will further challenge water resource management and disaster prevention for human society.

L406-412: Is this paragraph necessary?

As suggested, we have removed this paragraph.

L430-431: Do you mean ensemble mean precipitation whiplash characteristics?

As suggested, we have reworded this sentence.

The ensemble mean precipitation whiplash characteristics was adopted to show the projected forced changes, and differences between member responses represent internal climate variability of the results.

L431: How about “forced changes” instead of regimes?

As suggested, we have replaced "regimes" with "forced changes".

L449: Remove the word “defined”.

Removed.

L467: By model mean do you mean the ensemble mean?

As suggested, we have replaced "model mean" with "ensemble mean".

L467: Remove “weight equally”.

Removed.

L468-469: This sentence (the last sentence of the paragraph) is redundant.

As suggested, we have removed this sentence.

L471: How about “To compare with” instead of “For comparison with”?

As suggested, we have replaced "For comparison with" with "To compare with".

L472: Numbers of datasets (five, four, and four) in this sentence are not correct.

Corrected.

L476: Should be "The satellite-based dataset is".

Corrected.

L487: Instead of "annual-cycle-removed", try "annual cycle of the is removed."

As suggested, we have reworded this sentence.

To standardize the data and eliminate the effects of precipitation seasonality, the annual cycle of the time series of precipitation totals at the sub-seasonal scale is removed.

L495: "conclusions are not".

Corrected.

L510-512: What do you mean here by "does not overlap temporally"?

According to our principle of considering 30 days of cumulative precipitation, for example, if a dry extreme occurs in June 30 (implying that the cumulated precipitation status from June 1 to June 30 is dry), while it shifts to wet extremes in July 18 (implying that the cumulated precipitation status from June 18 to July 18 is wet), the temporal overlap from June 18 to June 30 implies a rapid transition between dry and wet conditions. If the cumulative precipitation of two opposite events does not overlap temporally, it implies that there is a slow recovery process from the dry (wet) extreme event to normal before the shift to the wet (dry) extreme event. We exclude such events from the rapid transition.

L548: "corresponds to December 31"

Corrected.

L562: Relative change of what?

Reworded.

L575: While it is the internal climate variability for CESM-LENS, for CMIP6, the noise includes the intermodel variability as well. This should be noted here.

As suggested, we have noted this point in this paragraph. In addition, we have checked all relevant description throughout the main text and supplementary materials.

L586: Change to "... extremes, whiplash, and precipitation totals"

Corrected.

L585-591: This is mentioned in the figure 4 caption. Do you need this paragraph?

As suggested, we have removed this paragraph.

L592: Change to "Anthropogenic influence on precipitation whiplash"

Changed.

L613: Removed "from the study days".

Removed.

L623: SAsiaM is missing the second "s" throughout the manuscript.

Corrected.

L890: ")" is missing?

Corrected.

L897: The figure title says "transition duration". It should be changed to the occurrence frequency.

Corrected.

L911: This figure caption is not clear. Please consider rewriting it.

As suggested, we have rewritten this figure caption.

L916: Use darker instead of deeper.

Corrected.

L918: In the figure legends, Instead of "fewer E", how about "less E"?

Considering that the word "event" is a countable noun, we prefer to use "fewer E".

L934: Replace "Average atmospheric anomalies".

We have replaced “Average” with “Ensemble mean”.

Reviewer #2

Recommendation: Accept after minor revisions

General comments:

The authors carefully addressed the concerns that were raised in my initial review and revised the manuscript for improvement. I think the revised manuscript is much improved and is worthy of publication in Nature Communications. However, I have found some typos as listed below, requiring careful checking of typos throughout the entire revised manuscript before submitting it.

Thank you very much for your time involved in reviewing the manuscript and your very encouraging comments. We have carefully and thoroughly proofread the manuscript to correct all the grammar and typos. The response to each point raised is given alongside the relevant specific comment below. Throughout, *reviewer comments* are in *blue* font and *italic* type, and our point-by-point response is in black font.

Minor revisions recommended:

1. [L472] *It does not seem that this study use “five reanalysis, four satellite-based datasets and four ground-based datasets”. Please correct this.*

Corrected.

2. [L298] *“Figs. 6c-f” might be “Figs. 6d-f”.*

Corrected.

3. [L298] *“Supplementary Figs. 21c-f” might be “Supplementary Figs. 21d-f”.*

Corrected.

4. [L530] *“Annual-cycle-removed cumulative precipitation totals” might be “Standardized annual-cycle-removed cumulative precipitation anomalies”.*

Corrected.

5. [L555] *“n is the total amount of whiplash at that grid” might be “n is the number of total years”.*

In Equation 6, i indicates a whiplash event, thus n indicates the total amount of whiplash at that grid. In this study, both the CESM-LENS and CMIP6 simulations consist of 365 days per year (with February 29 excluded for consistency). As a result, $\bar{m} = 365$ regardless of the number of events.

Reviewer #3

My main requests have been addressed and the manuscript has been substantially improved compared to the original draft. I have therefore no more objection to the publication of the paper in Nature Communications.

We sincerely thank you for your extensive and thoughtful comments which have helped a lot in improving our manuscript in the first round of revision, and we sincerely thank you for this great opportunity to improve our study. We have benefited greatly during the revision process of the manuscript.